# NETS: A Non-Equilibrium Transport Sampler

**Michael S. Albergo** [1] [2]   **Eric Vanden-Eijnden** [3] [4]

## Abstract

We introduce the Non-Equilibrium Transport Sampler (NETS), an algorithm for sampling from unnormalized probability distributions. NETS builds on non-equilibrium sampling strategies that transport a simple base distribution into the target distribution in finite time, as pioneered in Neal's annealed importance sampling (AIS). In the continuous-time setting, this transport is accomplished by evolving walkers using Langevin dynamics with a time-dependent potential, while simultaneously evolving importance weights to debias their solutions following Jarzynski's equality. The key innovation of NETS is to add to the dynamics a learned drift term that offsets the need for these corrective weights by minimizing their variance through an objective that can be estimated without backpropagation and provably bounds the Kullback-Leibler divergence between the estimated and target distributions. NETS provides unbiased samples and features a tunable diffusion coefficient that can be adjusted after training to maximize the effective sample size. In experiments on standard benchmarks, high-dimensional Gaussian mixtures, and statistical lattice field theory models, NETS shows compelling performances.

## 1. Introduction

The aim of this paper is to sample probability distributions on $\mathbb{R}^d$ known only up to normalization. This problem, central to applications from statistical physics (Faulkner & Livingstone, 2023; Wilson, 1974; Hénin et al., 2022)

[1] Society of Fellows, Harvard University, Cambridge, MA, USA [2] NSF AI Institute for Artificial Intelligence and Fundamental Interactions, Cambridge, MA, USA [3] Machine Learning Lab, Capital Fund Management, Paris, France [4] Courant Institute of Mathematical Sciences, New York, NY, USA. Correspondence to: Michael S. Albergo <malbergo@fas.harvard.edu>, Eric Vanden-Eijnden <eve2@cims.nyu.edu>.

*Proceedings of the 42nd International Conference on Machine Learning*, Vancouver, Canada. PMLR 267, 2025. Copyright 2025 by the author(s).

to Bayesian inference (Neal, 1993), becomes particularly challenging for non-log-concave targets. In such cases, traditional ergodic sampling methods based on Markov Chain Monte Carlo (MCMC) or Langevin dynamics often exhibit prohibitively slow convergence rates.

An alternative approach offered by non-equilibrium sampling strategies is to transport samples on non-stationary distributions that evolve from a simple base distribution (e.g., a normal distribution) to the target in finite time. Methods like Annealed Importance Sampling (AIS) (Neal, 2001), Sequential Monte Carlo (SMC) (Del Moral, 1997; Doucet et al., 2001), and their continuous-time variants based on Jarzynski's equality (Hartmann et al., 2018) achieve this transport through a dynamical quench, using importance weights to correct for the bias arising when the walkers' distribution lags behind their evolving target. However, these methods can fail when the lag is too strong, leading to high-variance weights.

We address this limitation by modifying the dynamics via some additional transport learned to guide toward the target. In the context of quenches performed via Langevin dynamics on an evolving potential, we show that an optimal drift can be added to eliminate the need for importance weights entirely, and can be characterized as the minimizer of objective functions amenable to empirical estimation. This leads to an unbiased sampling strategy where importance weights, while still available for exact correction, have substantially reduced variance due to the improved transport.

In sum, our work makes the following **main contributions**:

- We introduce the Non-Equilibrium Transport Sampler (NETS), which augments Langevin dynamics on an evolving potential with learned transport. We show that the method remains unbiased through a generalization of Jarzynski's equality, and that it can be used in concert with sequential Monte-Carlo (SMC) strategies.

- We demonstrate that the optimal drift for this additional transport minimizes an objective function, based on physics-informed neural networks (PINN), whose gradient can be estimated without backpropagating through the sampling equations.

- We show that the PINN objective has two key properties:

it is off-policy (requiring no samples from the target density) and it bounds the KL-divergence between the sampler and target distributions.

- We discuss other possible objectives to make connections with Action Matching (AM) as well as Controlled Monte Carlo Diffusions (CMCD).

- We establish that NETS's performance can be optimized post-training by tuning both the integration time-step and diffusion coefficient, which we demonstrate through high-dimensional numerical experiments.

### 1.1. Related work

**Dynamical Measure Transport.**  Modern generative models for continuous data often employ dynamical measure transport, where samples from a base density are transformed into samples from a target density by solving ordinary or stochastic differential equations (ODE/SDE) with learnable drift coefficients. This approach, pioneered in (Chen et al., 2018; Grathwohl et al., 2019), has developed in several directions, notably with score-based diffusion models (Ho et al., 2020; Song et al., 2020) that frame drift estimation as quadratic regression. This formulation has led to more general frameworks (Albergo & Vanden-Eijnden, 2022; Lipman et al., 2022; Albergo et al., 2023; Liu et al., 2022; De Bortoli et al., 2021; Neklyudov et al., 2023). A key requirement of these methods is access to samples from both base and target distributions. Our work shows that similar models building on dynamical transport can be learned without prior access to data from the target.

**Augmenting sampling with learning.**  Augmenting MCMC and importance sampling proceures with transport has been an active area of research for the past decade. Early work makes use of the independence Metropolis algorithm (Hastings, 1970; Liu, 1996), in which proposals come from a transport map (Parno & Marzouk, 2018; Noé et al., 2019; Albergo et al., 2019; Gabrié et al., 2022) that are accepted or rejected based off their likelihood ratio with the target. These methods were further improved by combining them with AIS and SMC perspectives, learning incremental maps that connect a sequence of interpolating densities between the base and target (Arbel et al., 2021; Matthews et al., 2022; Midgley et al., 2023). Similar works in the high-energy physics community posit that interleaving stochastic updates within a sequence of maps can be seen as a form of non-equilibrium sampling (Caselle et al., 2022; Bonanno et al., 2024).

Following the success of generative models based on dynamical transport, several approaches have emerged to apply these ideas to sampling. Some translate concepts from diffusion models to minimize the KL divergence between model and target (Vargas et al., 2023; Berner et al., 2024),

while others reformulate sampling as a stochastic optimal control (SOC) problem (Zhang & Chen, 2021; Behjoo & Chertkov, 2024; Hua et al., 2024). However, these methods face a limitation: learning the optimal drift is a complicated optimization problem which becomes computationally prohibitive in high dimensions, and simplified approaches such as Akhound-Sadegh et al. (2024) introduce bias into their objective function. Alternative perspectives include using denoising oracles to simplify the sampling problem (Bruna & Han, 2024) and adapting graph-based distribution modeling techniques for sampling, including off-policy training (Malkin et al., 2023; Sendera et al., 2024).

Vargas et al. (2024) establish Controlled Monte Carlo Diffusions (CMCD), another unbiased sampler that, like ours, augments the dynamics with learned transport. The methods differ in how this drift is learned: CMCD derives a gradient-form objective through path integrals and Girsanov's theorem, requiring either backpropagation through the SDE or computation with a numerically unstable reference measure on a fixed grid. Our approach, based on Fokker-Planck equation manipulations, yields new objectives for the additional drift that avoid backpropagation entirely. Moreover, our optimize-then-discretize framework allows for post-training adaptation of both step size and time-dependent diffusion, providing tunable parameters to enhance performance. One of our objectives is a Physics-Informed Neural Network (PINN) loss, which has appeared elsewhere for sampling (Máté & Fleuret, 2023; Tian et al., 2024; Fan et al., 2024). We establish two key results: this objective is valid for annealed Langevin dynamics, and it directly controls both the KL divergence and the importance weights arising from Jarzynski's equality.

## 2. Methods

### 2.1. Setup and Notations

We assume that the target distribution is absolutely continuous with respect to the Lebesgue measure on $\mathbb{R}^d$, with probability density function (PDF) $\rho_1(x) = Z_1^{-1} e^{-U_1(x)}$: here $x \in \mathbb{R}^d$, $U : \mathbb{R}^d \to \mathbb{R}$ is a known energy potential, assumed twice differentiable and bounded below, and $Z_1 = \int_{\mathbb{R}^d} e^{-U_1(x)} dx < \infty$ is an unknown normalization constant, referred to as the partition function in physics and the evidence in statistics. Our aim is to generate samples from $\rho_1(x)$ so as to be able to estimate expectations with respect to this density. Additionally we wish to estimate $Z_1$.

To this end, we introduce a family of time-dependent potentials $U_t(x)$ that interpolate between a simple initial potential $U_0(x)$ (e.g., $U_0(x) = \frac{1}{2}|x|^2$) at $t = 0$ and the target potential $U_1(x)$ at $t = 1$. While linear interpolation, $U_t(x) = (1-t)U_0(x) + tU_1(x)$, provides a straightforward choice, more sophisticated interpolation schemes are often

preferable. Our only requirements are that $U_{t=0} = U_0$, $U_{t=1} = U_1$, and $U_t(x)$ is twice differentiable in $(t, x) \in [0, 1] \times \mathbb{R}^d$. We assume that the time-dependent PDF associated with this potential $U_t(x)$ is normalizable for all $t \in [0, 1]$ and denote it as

$$\rho_t(x) = Z_t^{-1} e^{-U_t(x)}, \quad Z_t = \int_{\mathbb{R}^d} e^{-U_t(x)} dx < \infty, \quad (1)$$

so that $\rho_{t=0}(x) = \rho_0(x)$ and $\rho_{t=1}(x) = \rho_1(x)$; we also assume that $\rho_0(x)$ is simple to sample (either directly or via MCMC or Langevin dynamics) and that its partition function $Z_0$ is known. To simplify the notations we introduce the free energy

$$F_t = -\log Z_t, \quad (2)$$

and note the useful identity

$$\partial_t F_t = \int_{R^d} \partial_t U_t(x) \rho_t(x) dx. \quad (3)$$

since $-\partial_t \log \int_{\mathbb{R}^d} e^{-U_t} = \int_{\mathbb{R}^d} \partial_t U_t e^{-U_t} / \int_{R^d} e^{-U_t}$.

## 2.2. Non-equilibrium Sampling with Importance Weights

Annealed importance sampling (AIS) uses a finite sequence of MCMC moves that satisfy detailed-balance locally in time but not globally, thereby introducing a bias that can be corrected with weights. Here we use a time-continuous variant of AIS based on Jarzynski equality.

By definition of the PDF in (1), $\nabla \rho_t(x) = -\nabla U_t(x) \rho_t(x)$ and hence, for any $\epsilon_t \geq 0$, we have

$$0 = \epsilon_t \nabla \cdot (\nabla U_t \rho_t + \nabla \rho_t). \quad (4)$$

Since we also have

$$\partial_t \rho_t = -(\partial_t U_t - \partial_t F_t) \rho_t, \quad (5)$$

we can combine these last two equations to deduce that

$$\partial_t \rho_t = \epsilon_t \nabla \cdot (\nabla U_t \rho_t + \nabla \rho_t) - (\partial_t U_t - \partial_t F_t) \rho_t. \quad (6)$$

The effect of the last term at the right hand-side of this equation can be accounted for by using weights. To see how, extend the phase space to $(x, a) \in \mathbb{R}^{d+1}$ and introduce the PDF $f_t(x, a)$ solution to the Fokker-Planck equation (FPE)

$$\partial_t f_t = \epsilon_t \nabla \cdot (\nabla U_t f_t + \nabla f_t) + \partial_t U_t \partial_a f_t, \quad (7)$$

with initial condition $f_{t=0}(x, a) = \delta(a) \rho_0(x)$. A direct calculation using (3) (for details see Appendix A) shows that

$$\rho_t(x) = \frac{\int_{\mathbb{R}} e^a f_t(x, a) da}{\int_{\mathbb{R}^{d+1}} e^a f_t(y, a) dady}. \quad (8)$$

Therefore we can use the solution to the SDE associated with the FPE (7) in the extended space to estimate expectations with respect to $\rho_t(x)$:

**Proposition 2.1** (Jarzynski equality). *Let $(X_t, A_t)$ solve the coupled system of SDE/ODE*

$$dX_t = -\epsilon_t \nabla U_t(X_t) dt + \sqrt{2\epsilon_t} dW_t, \quad X_0 \sim \rho_0, \quad (9)$$
$$dA_t = -\partial_t U_t(X_t) dt, \quad A_0 = 0, \quad (10)$$

*where $\epsilon_t \geq 0$ is a time-dependent diffusion coefficient and $W_t \in \mathbb{R}^d$ is the Wiener process. Then for all $t \in [0, 1]$ and any test function $h : \mathbb{R}^d \to \mathbb{R}$, we have*

$$\int_{\mathbb{R}^d} h(x) \rho_t(x) dx = \frac{\mathbb{E}[e^{A_t} h(X_t)]}{\mathbb{E}[e^{A_t}]}, \quad (11)$$
$$Z_t / Z_0 = e^{-F_t + F_0} = \mathbb{E}[e^{A_t}]$$

*where the expectations are taken over the law of $(X_t, A_t)$.*

The proof of this proposition is given in Appendix A and it relies on the identity $\int_{\mathbb{R}^d} h(x) \rho_t(x) = \int_{\mathbb{R}^{1+d}} e^a h(x) f_t(x, a) dadx / \int_{\mathbb{R}^{d+1}} f_t(x, a) dadx$ which follows from (8). The second equation in (11) for the free energy $F_t$ is what is referred to as Jarzynski's equality, and was originally surmised in the context of non-equilibrium thermodynamics (Jarzynski, 1997).

**Remark 2.2.** We stress that it is key to use the weights $e^{A_t}$ in (11) because $\rho_t(x)$ *is not the PDF of $X_t$ in general*. Indeed, if we denote by $\tilde{\rho}_t(x)$ the PDF of $X_t$, it satisfies

$$\partial_t \tilde{\rho}_t = \epsilon_t \nabla \cdot (\nabla U_t \tilde{\rho}_t + \nabla \tilde{\rho}_t), \quad \tilde{\rho}_{t=0} = \rho. \quad (12)$$

This FPE misses the term $-(\partial_t U_t - \partial_t F_t)\rho_t$ at the right hand-side of (6), and as a result $\tilde{\rho}_t(x) \neq \rho_t(x)$ in general – intuitively, $\tilde{\rho}_t(x)$ lags behind $\rho_t(x)$ when the potential $U_t(x)$ evolves and this lag is what the weights in (11) correct for.

It is important to realize that, while (11) is exact, estimators based on this relation can be high variance if the lag between the PDF $\tilde{\rho}_t(x)$ of $X_t$ and $\rho_t(x)$ is too pronounced. This issue can be alleviated by using resampling methods as is done in SMC (Doucet et al., 2001). Here we will address it by adding some additional drift in (9) that will reduce the lag and as a result lower the variance of the weights.

## 2.3. Non-equilibrium Sampling with Perfect Transport

We can add a transport term to eliminate the need of the weights. To see how, let $b_t(x) \in \mathbb{R}^d$ be a velocity field which at all times $t \in [0, 1]$ satisfies

$$\nabla \cdot (b_t \rho_t) = -\partial_t \rho_t. \quad (13)$$

We stress that this is an equation for $b_t(x)$ in which $\rho_t(x)$ is prescribed and given by (1): In Appendix C we show how to express the solution to (13) via Feynman-Kac formula. If (13) is satisfied, then we can combine this equation with (4) and (5) to arrive at

$$\partial_t \rho_t = \epsilon_t \nabla \cdot (\nabla U_t \rho_t + \nabla \rho_t) - \nabla \cdot (b_t \rho_t), \quad (14)$$

which is a standard FPE. Therefore the solution to the SDE associated with (14) allows us to sample $\rho_t(x)$ directly (without weights). We phrase this result as:

**Proposition 2.3** (Sampling with perfect additional transport.)**.** *Let $b_t(x)$ be a solution to (13) and let $X_t^b$ satisfy the SDE*

$$dX_t^b = -\epsilon_t \nabla U_t(X_t^b)dt + b_t(X_t^b)dt + \sqrt{2\epsilon_t}dW_t, \quad (15)$$

*with $X_0^b \sim \rho_0$ and where $\epsilon_t \geq 0$ is a time-dependent diffusion coefficient. Then $\rho_t(x)$ is the PDF of $X_T^b$, i.e. for all $t \in [0, 1]$ and, given any test function $h : \mathbb{R}^d \to \mathbb{R}$, we have*

$$\int_{\mathbb{R}^d} h(x)\rho_t(x)dx = \mathbb{E}[h(X_t^b)], \quad (16)$$

*where the expectation at the right-hand side is taken over the law of $(X_t^b)$.*

This proposition is proven in Appendix A and it shows that we can in principle get rid of the weights altogether by adding the drift $b_t(x)$ in the Langevin SDE. This possibility was first noted in Vaikuntanathan & Jarzynski (2008) and is also exploited in Tian et al. (2024) for deterministic dynamics (i.e. setting $\epsilon_t = 0$ in (15)) and in Vargas et al. (2024) using the SDE (15). Of course, in practice we need to estimate $b_t(x)$, and also correct for sampling errors if this drift is imperfectly learned. Let us discuss this second question first, and defer the derivation of objectives to learn $b_t(x)$ to Secs. 2.5 and E. In Appendix C we show how to express the solution to (13) via Feynman-Kac formula.

## 2.4. Non-Equilibrium Transport Sampler

Let us now combine the approaches discussed in Secs. 2.2 and 2.3 to design samplers in which we use an added transport, possibly imperfect, and importance weights.

To this end, suppose that we wish to add an additional transport term $-\nabla \cdot (\hat{b}_t \rho_t)$ in (6), where $\hat{b}_t(x) \in \mathbb{R}^d$ is some given velocity that does not necessarily solve (13). Using the expression in (1) for $\rho_t(x)$, we have the identity

$$-\nabla \cdot (\hat{b}_t \rho_t) = -\nabla \cdot \hat{b}_t \rho_t + \nabla U_t \cdot \hat{b}_t \rho_t \quad (17)$$

Therefore we can rewrite (4) equivalently as

$$\partial_t \rho_t = \epsilon_t \nabla \cdot (\nabla U_t \rho_t + \nabla \rho_t) - \nabla \cdot (\hat{b}_t \rho_t) \\ + (\nabla \cdot \hat{b}_t - \nabla U_t \cdot \hat{b}_t - \partial_t U_t + \partial_t F_t)\rho_t \quad (18)$$

We can now proceed as we did with (4) and extend state space to account for the effect of the terms $(\nabla \cdot \hat{b}_t - \nabla U_t \cdot \hat{b}_t - \partial_t U_t + \partial_t F_t)\rho_t$ in this equation via weights, while having the term $-\nabla \cdot (\hat{b}_t(x)\rho_t(x))$ contribute to some additional transport. This leads us to a result originally obtained in Vaikuntanathan & Jarzynski (2008) and recently re-derived in Vargas et al. (2024):

**Proposition 2.4** (Nonequilibrium Transport Sampler (NETS))**.** *Let $(X_t^{\hat{b}}, A_t^{\hat{b}})$ solve the coupled system of SDE/ODE*

$$dX_t^{\hat{b}} = -\epsilon_t \nabla U_t(X_t^{\hat{b}})dt + \hat{b}_t(X_t^{\hat{b}})dt + \sqrt{2\epsilon_t}dW_t, \quad (19)$$

$$dA_t^{\hat{b}} = \nabla \cdot \hat{b}_t(X_t^{\hat{b}})dt - \nabla U_t(X_t^{\hat{b}}) \cdot \hat{b}_t(X_t^{\hat{b}})dt \\ - \partial_t U_t(X_t^{\hat{b}})dt, \quad (20)$$

*with $X_0^{\hat{b}} \sim \rho_0$ and $A_0^{\hat{b}} = 0$, and where $\epsilon_t \geq 0$ is a time-dependent diffusion coefficient. Then for all $t \in [0, 1]$ and any test function $h : \mathbb{R}^d \to \mathbb{R}$, we have*

$$\int_{\mathbb{R}^d} h(x)\rho_t(x)dx = \frac{\mathbb{E}[e^{A_t^{\hat{b}}}h(X_t^{\hat{b}})]}{\mathbb{E}[e^{A_t^{\hat{b}}}]}, \quad (21)$$

$$Z_t/Z_0 = e^{-F_t+F_0} = \mathbb{E}[e^{A_t^{\hat{b}}}].$$

*where the expectations are taken over the law of $(X_t^{\hat{b}}, A_t^{\hat{b}})$.*

A proof of this proposition using manipulations of the FPE is given in Appendix A, which will allow us to write down a variety of new loss functions for learning $\hat{b}_t$; for an alternative proof using Girsanov theorem, see Vargas et al. (2024). For completeness, in Appendix B we also give a time-discretized version of Proposition 2.4, and in Appendix D we generalize it in two ways: to include inertia and to turn $t$ into a vector coordinate for multimarginal sampling. We also discuss the connection between NETS and the method of Vargas et al. (2024) in Appendix F.

Notice that, if $\hat{b}_t(x) = 0$, the equations in Proposition 2.4 simply reduce to those in Proposition 2.1, whereas if $\hat{b}_t(x) = b_t(x)$ solves (13) we can show that

$$A_t^b = -F_t + F_0, \quad (22)$$

i.e. the weights have zero variance and give the free energy difference. Indeed, by expanding both sides of (13) and dividing them by $\rho_t(x) > 0$, this equation can equivalently be written as

$$\nabla \cdot b_t - \nabla U_t \cdot b_t = \partial_t U_t - \partial_t F_t. \quad (23)$$

As a result, when $\hat{b}_t(x) = b_t(x)$, (20) reduces to

$$dA_t^b = -\partial_t F_t dt, \qquad A_0^b = 0, \quad (24)$$

and the solution to this equation is (22). In practice, achieving zero variance of the weights by estimating $b_t(x)$ exactly is not generally possible, but having a good approximation of $b_t(x)$ can help reducing this variance dramatically, as we will illustrate below via experiments.

## 2.5. Estimating the Drift $b_t(x)$ via a PINN Objective

Equation (23) can be used to derive an objective for both $b_t(x)$ and $F_t$. The reason is that in this equation the unknown $\partial_t F_t$ can be viewed as factor that guarantees solvability: indeed, integrating both sides of (13) gives $0 =$

$-\partial_t \int_{\mathbb{R}^d} U_t(x)\rho_t(x)dx + \partial_t F_t$, which, by (3), is satisfied if and only if $F_t$ is (up to a constant fixed by $F_0 = -\log Z_0$) the exact free energy (2). This offers the possibility to learn both $b_t(x)$ and $F_t$ variationally using an objective fitting the framework of physics informed neural networks (PINNs):

**Proposition 2.5** (PINN objective). *Given any $T \in (0, 1]$ and any PDF $\hat{\rho}_t(x) > 0$ consider the objective for $(\hat{b}, \hat{F})$ given by:*

$$L_{PINN}^T[\hat{b}, \hat{F}] = \int_0^T \mathbb{E}\big[|q_t(x_t)|^2\big]dt, \qquad (25)$$

*where $x_t \sim \hat{\rho}_t$ and we denote*

$$q_t(x) = \nabla \cdot \hat{b}_t(x) - \nabla U_t(x) \cdot \hat{b}_t(x) - \partial_t U_t(x) + \partial_t \hat{F}_t. \quad (26)$$

*Then $\min_{\hat{b}, \hat{F}} L_{PINN}^T[\hat{b}, \hat{F}] = 0$, and all minimizers $(b, F)$ are such that and $b_t(x)$ solves (23) and $F_t$ is the free energy (2) for all $t \in [0, T]$.*

This result is proven in Appendix A: in practice, we will use $T \in (0, 1]$ for annealing but ultimately we are interested in the result when $T = 1$. Note that since the expectation over an arbitrary $\hat{\rho}_t(x)$ in (25), it can be used as an off-policy objective. It is however natural to use $\hat{\rho}_t(x) = \rho_t(x)$ since it allows us to put statistical weight in the objective precisely in the regions where we need $b_t(x)$ to transport probability mass. In either case, there is no need to backpropagate through simulation of the SDE used to produce data. We show in Section 2.7 below how the expectation over $\rho_t(x)$ can be estimated without bias to arrive at an empirical estimator for (25) when $\hat{\rho}_t(x) = \rho_t(x)$. Note also that, while minimization of the objective (25) gives an estimate $\hat{F}_t$ of the free energy, it is not needed at sampling time when solving (19). An objective similar to (25) was also recently posited in Máté & Fleuret (2023); Tian et al. (2024) for use with deterministic flows. Here, we devise it in the context of augmenting annealed Langevin dynamics.

In addition to the above results, in Appendix D we supply extensions to the setup to the case where there are multiple marginals and where the stochastic dynamics have inertia. We also discuss an alternative objective, relying on the action matching formalism (Neklyudov et al., 2023), in Appendix E.

## 2.6. Control of the Kullback-Leibler Divergence

One advantage of the PINN objective (25) is that we know that its minimum is zero, and hence we can track its value to monitor convergence when minimizing (25) by gradient descent, as we do below. Another advantage of the loss (25) is that it controls the quality of the transport as measured by the Kullback-Leibler divergence:

**Proposition 2.6** (KL control). *Let $\hat{\rho}_t$ be the solution to the*

*Fokker-Planck equation*

$$\partial_t \hat{\rho}_t + \nabla \cdot (\hat{b}_t \hat{\rho}_t) = \epsilon_t \nabla \cdot (\nabla U_t \hat{\rho}_t + \nabla \hat{\rho}_t), \qquad (27)$$

*with $\hat{\rho}_{t=0} = \rho_0$ and where $\hat{b}_t(x)$ is some predefined velocity field and $\epsilon_t \geq 0$. Then, we have*

$$D_{KL}(\hat{\rho}_{t=1}||\rho_1) \leq \sqrt{L_{PINN}^{T=1}(\hat{b}, F)}. \qquad (28)$$

*where $F_t$ is the free energy. In addition, given any estimate $\hat{F}_t$ such that $\int_0^1 |\partial_t \hat{F}_t - \partial F_t|^2 dt \leq \delta$ for some $\delta \geq 0$, we have*

$$D_{KL}(\hat{\rho}_{t=1}||\rho_1) \leq \sqrt{2L_{PINN}^{T=1}(\hat{b}, \hat{F}) + 2\delta}. \qquad (29)$$

This proposition is proven in Appendix A. Notice that the bound (28) can be estimated by using $\partial_t F_t = \mathbb{E}[e^{A_t^{\hat{b}}} \partial_t U_t(X_t^{\hat{b}})]/\mathbb{E}[e^{A_t^{\hat{b}}}]$ in the PINN loss (25).

## 2.7. Implementation

If we minimize (25) *off-policy*, i.e. with samples $x_t$ from some $\hat{\rho}_t \neq \rho_t$, this is perfectly valid, but may be inefficient for learning $\hat{b}_t$ over the support necessary for the problem. If we decide instead to set $\hat{\rho}_t(x) = \rho_t(x)$, since the SDEs in (19) and (20) can be used with any $\hat{b}_t(x)$ to estimate expectation over $\rho_t(x)$ via (21), we can write the PINN objective on-policy as

$$L_{\text{PINN}}^T[\hat{b}, \hat{F}] = \int_0^T \frac{\mathbb{E}\big[e^{A_t^{\hat{b}}}|q_t(X_t^{\hat{b}})|^2\big]}{\mathbb{E}\big[e^{A_t^{\hat{b}}}\big]}dt \qquad (30)$$

These expectations can be estimated empirically over a population of solutions to (19) and (20). Crucially, since we can switch from off-policy to on-policy after taking the gradient of the PINN objective, *when computing the gradient of (30) over $\hat{b}_t(x)$, $(X_t^{\hat{b}}, A_t^{\hat{b}})$ can be considered independent of $\hat{b}_t(x)$ and do not need to be differentiated over.* In other words, the method does not require backpropagation through the simulation even if used on-policy, i.e. even though it uses the current value of $\hat{b}_t$ to estimate the loss and its gradient. Finally note that we can use the ODE (20) for $A_t^{\hat{b}}$ to write (30) as

$$L_{\text{PINN}}^T[\hat{b}, \hat{F}] = \int_0^T \frac{\mathbb{E}\big[e^{A_t^{\hat{b}}}|\partial_t A_t^{\hat{b}} + \partial_t \hat{F}_t|^2\big]}{\mathbb{E}\big[e^{A_t^{\hat{b}}}\big]}dt \qquad (31)$$

Since $\mathbb{E}[e^{A_t^{\hat{b}}} \partial_t A_t^{\hat{b}}]/\mathbb{E}[e^{A_t^{\hat{b}}}] = \partial_t \log \mathbb{E}[e^{A_t^{\hat{b}}}] = -\partial_t F_t$, (31) clearly shows that this loss controls the variance of $\partial_t A_t^{\hat{b}}$, which directly connects the Jarzynski weights to the PINN objective.

The computation of the divergence $\nabla \cdot b_t(x)$ in the PINN objective given in (25) can be avoided by using Hutchinson's trace estimator, see Appendix G.

**Algorithm 1** Training: Note that the resultant set of walkers across time slices $\{x_k^i\}$ are detached from the computational graph when taking a gradient step (*off-policy learning*).

1: **Initialize:** $n$ walkers, $x_0 \sim \rho_0$, $A_0 = 0$, $K$ time steps, model parameters for $\{\hat{b}_t, \hat{F}_t\}$, diffusion coefficient $\epsilon_t$, learning rate $\eta$
2: **repeat**
3:     Randomize time grid:    $t_0, t_1, \ldots, t_K$    $\sim$ Uniform$(0, T)$, sort such that $t_0 < t_1 < \cdots < t_K$
4:     **for** $k = 0, \ldots, K$ **do**
5:        $\Delta t_k = t_{k+1} - t_k$,
6:        **for** each walker $i = 1, \ldots, n$ **do**
7:           $x_{k+1}^i = x_k^i + [\hat{b}_{t_k}(x_k^i) - \epsilon_{t_k} \nabla U_{t_k}(x_k^i)] \Delta t_k$
               $+\sqrt{2\epsilon_{t_k}}(W_{t_{k+1}}^i - W_{t_k}^i)$
8:           $A_{k+1}^i = A_k^i + [\nabla \cdot \hat{b}_{t_k}(x_k^i) - \partial_t U_{t_k}(x_k^i)$
               $-\hat{b}_{t_k}(x_k^i) \cdot \nabla U_{t_k}(x_k^i)] \Delta t_k$
9:     Estimate (30) by replacing the expectation by an empirical average over the $n$ walkers and the time integral by an empirical average over $t_0, \ldots, t_K$.
10:     Take gradient descent step to update the model parameters.
11: **until** converged

Learning $b_t(x)$ and $F_t$ for $t \in [0, 1]$ from the start can be challenging if the initial $\hat{b}_t(x)$ is far from exact and the weights gets large variance as $t$ increases. This problem can be alleviated by estimating $b_t(x)$ sequentially. In practice, this amounts to annealing $T$ from a small initial value to $T = 1$, in such a way that $b_t(x)$ is learned sufficiently accurately so that variance of the weights remains small. This variance can be estimated on the fly, which also give us an estimate of the effective sample size (ESS) of the population at all times $t \in [0, 1]$.

Note that we can also employ resampling strategies of the type used in SMC to keep the variance of the weights low (Doucet et al., 2001; Bolić et al., 2004).

Details for the numerical implementation of the minimization of the objective (25) is summarized in Algorithm 1.

### 2.8. Learning the $U_t$ of Stochastic Interpolants

The choice of the potential $U_t$ used in the annealing can have a significant impact on both the learnability of $b_t$ and the numerical stability of solving (19). A desirable characteristic is that $U_t$ gives a density $\rho_t$ that is geometrically smooth in its evolution between $\rho_0$ and $\rho_1$, so that transport via vector fields is simple. One approach toward achieving this is to use the drift associated with a stochastic interpolant (Albergo & Vanden-Eijnden, 2022; Lipman et al., 2022), i.e., the stochastic process defined as

$$I_t = \alpha_t x_0 + \beta_t x_1, \quad x_0 \sim \rho_0, \quad x_1 \sim \rho_1, \quad (32)$$

where the coefficients $\alpha_t, \beta_t$ satisfy $\alpha_0 = \beta_1 = 1$, $\alpha_1 = \beta_0 = 0$ and $\dot{\alpha}_t < 0$, $\dot{\beta}_t > 0$. The PDF $\rho_t$ of $I_t$ satisfies (13) with a drift $b_t$ given by

$$b_t(x) = \mathbb{E}[\dot{\alpha}_t x_0 + \dot{\beta}_t x_1 \mid I_t = x], \quad (33)$$

where the expectation is taken over the law of $x_0, x_1$ conditional on $I_t = x$. When $\rho_0 := \mathcal{N}(0, I_d)$, an application of Stein's lemma indicates that the gradient of the potential $U_t$ associated with $\rho_t$ is given by

$$\nabla U_t(x) = \alpha_t^{-1} \mathbb{E}[x_0 \mid I_t = x]. \quad (34)$$

While theoretically appealing, these relations are not immediately useful for two reasons. First, this velocity field can only be regressed when samples from $\rho_1$ are readily available, which is not the case in our setting. Second, the potential $U_t$ associated with this $\rho_t$ is not analytically known. However, by combining (33) with (34) and using $x = \mathbb{E}[I_t \mid I_t = x]$, we notice that $b_t$ can be written as

$$b_t(x) = (\dot{\alpha}_t \alpha_t - \alpha_t^2 \dot{\beta}_t \beta_t^{-1}) \nabla U_t(x) + \dot{\beta}_t \beta_t^{-1} x, \quad (35)$$

which we can exploit to directly learn the $U_t$ associated with the interpolant and, in the process, solve the transport problem. A convenient choice is to set, for example, $\alpha_t = \sqrt{1 - t^2}$ and $\beta_t = t$, so that (35) reduces to

$$t b_t(x) = x - \nabla U_t(x). \quad (36)$$

If we parameterize the potential as $U_t = \hat{U}_t^f$ with

$$\hat{U}_t^f(x) = (1 - t)\tfrac{1}{2}|x|^2 + t U_1(x) + t(1 - t)\hat{f}_t(x), \quad (37)$$

where $\hat{f}_t(x) : [0, 1] \times \mathbb{R}^d \to \mathbb{R}$ is a neural network, (36) gives us an expression for the drift also in terms of $\hat{f}_t$:

$$\begin{aligned} \hat{b}_t^f(x) &= t^{-1}(x - \nabla \hat{U}_t^f(x)) \\ &= x - \nabla U_1(x) - (1 - t)\nabla \hat{f}_t(x). \end{aligned} \quad (38)$$

This allows us to write the PINN objective (25) as an objective for $(\hat{f}, \hat{F})$:

$$L_{\text{PINN}}[\hat{f}, \hat{F}] = \int_0^1 \mathbb{E}[|q_t^f(x_t)|^2] \, dt, \quad (39)$$

where $q_t^f(x_t)$ is obtained from (26) by replacing $U_t$ with (37) and $b_t$ with (38):

$$q_t^f(x) = \nabla \cdot \hat{b}_t^f(x) - \nabla \hat{U}_t^f(x) \cdot \hat{b}_t^f(x) - \partial_t \hat{U}_t^f(x) + \partial_t \hat{F}_t. \quad (40)$$

When minimized to zero, the objective (39) yields both a drift $\hat{b}_t^f$ and a potential $\hat{U}_t^f$ such that the resulting density $\rho_t = e^{-\hat{U}_t^f + \hat{F}_t}$ matches the PDF of the interpolant $I_t$ and possesses favorable transport properties.

| GMM ($d = 2$) | | |
|---|---|---|
| **Algorithm** | **ESS ↑** | $\mathcal{W}_2$ ↓ |
| FAB | $0.653 \pm 0.017$ | $12.0 \pm 5.73$ |
| PIS | $0.295 \pm 0.018$ | $7.64 \pm 0.92$ |
| DDS | $0.687 \pm 0.208$ | $9.31 \pm 0.82$ |
| pDEM | $0.634 \pm 0.084$ | $12.20 \pm 0.14$ |
| iDEM | $0.734 \pm 0.092$ | $7.42 \pm 3.44$ |
| CMCD-KL | $0.268 \pm 0.069$ | $9.32 \pm 0.71$ |
| CMCD-LV | $0.655 \pm 0.023$ | $4.01 \pm 0.25$ |
| NETS-AM $\epsilon_t = 5$ (ours) | $0.808 \pm 0.031$ | $3.89 \pm 0.22$ |
| NETS-PINN $\epsilon_t = 0$ (ours) | $\mathbf{0.954 \pm 0.003}$ | $3.55 \pm 0.57$ |
| NETS-PINN $\epsilon_t = 4$ (ours) | $\mathbf{0.979 \pm 0.002}$ | $\mathbf{3.14 \pm 0.46}$ |
| NETS-PINN-resample (ours) | $\mathbf{0.993 \pm 0.004}$ | $\mathbf{3.27 \pm 0.31}$ |

*Table 1.* Performance of NETS in terms of ESS and $\mathcal{W}_2$ metrics for 40-mode GMM ($d = 2$) with comparative results quoted from Akhound-Sadegh et al. (2024) for reproducibility.

## 3. Numerical Experiments

In what follows, we test the NETS method, for both the PINN objective (25) and the action matching objective (119), on standard challenging sampling benchmarks. We then study how the method scales in comparison to baselines, particularly AIS on its own, by testing it on an increasingly high dimensional Gaussian Mixture Models (GMM). Following that, we show that it has orders of magnitude better statistical efficiency as compared to AIS on its own when applied to the study of lattice field theories, even past the phase transition of these theories and in 400 dimensions (an $L = 20 \times L = 20$ lattice).

### 3.1. 40-Mode Gaussian Mixture

A common benchmark for machine learning augmented samplers that originally appeared in the paper introducing Flow Annealed Importance Sampling Bootstrap (FAB) (Midgley et al., 2023) is a 40-mode GMM in 2-dimensions for which the means of the mixture components span from $-40$ to $40$. The high variance and many wells make this problem challenging for re-weighting or locally updating MCMC processes. We choose as a time dependent potential $U_t(x)$ that linearly interpolates the means of the GMM with $U_0(x)$ the potential for a standard multivariate Gaussian with standard deviation scale $\sigma = 2$.

We train a simple feed-forward neural network of width 256 against both the PINN objective (25), parameterizing $(\hat{b}, \hat{F})$, or the action matching objective (119), parameterizing $\hat{\phi}$. We compare the learned model from both objectives to recent related literature: FAB, Path Integral Sampler (PIS) (Zhang & Chen, 2021), Denoising Diffusion Sampler (DDS) (Vargas et al., 2023), and Denoising Energy Matching (pDEM, iDEM) (Akhound-Sadegh et al., 2024). For reproducibility with the benchmarks provided in the latter method, we compute the effective sample size (ESS) estimated from 2000 generated samples as well as the $2-$Wasserstein ($\mathcal{W}_2$) distance between the model and the target. As noted in Table 1, all proposed variants of

NETS outperform existing methods. In addition, because our method can be turned into an SMC method by including resampling during the generation, we can push the acceptance rate of the same learned PINN model to nearly 100% by using a single resampling step when the ESS of the walkers dropped below 98%. NETS uses 100 sampling steps and an $\epsilon_t = 0.0, 4.0$ in the SDE. Note that NETS and CMCD as published use different interpolating potentials on this example based on code conventions, but they could be the same.

### 3.2. Funnel and Student-t Mixture

We next test NETS on Neal's funnel, a challenging synthetic target distribution which exhibits correlations at different scales across its 10 dimensions, as well as the 50-dimensional Mixture of Student-T (MoS) distribution used in (Blessing et al., 2024). The definitions of the target densities and the interpolating potentials are given in Appendix H. Heuristically, the first dimension is Gaussian with variance $\sigma^2 = 9$, and the other 9 dimensions are conditionally Gaussian with variance $\exp(x_0)$, creating the funnel.

We again parameterize $(\hat{b}, \hat{F})$ or $\hat{\phi}$ using simple feed-forward neural networks, this time of hidden size 512. We use 100 sampling steps for both, with diffusion coefficients given in the caption of Table 2. Following (Blessing et al., 2024), we compute the maximum mean discrepancy (MMD) and $\mathcal{W}_2$ distance between 2000 samples from the model and 2000 samples from the target and compare to related methods in Table 2. NETS outperforms other methods with both losses on the high dimensional MoS target in both metrics. In addition this can be improved using SMC-style resampling in the interpolation when the ESS drops below 70%. NETS matches the best performance in MMD for the Funnel distribution, but it is slightly worse in $\mathcal{W}_2$.

| Algorithm | Funnel ($d = 10$) | | MoS ($d = 50$) | |
|---|---|---|---|---|
| | **MMD ↓** | $\mathcal{W}_2$ ↓ | **MMD ↓** | $\mathcal{W}_2$ ↓ |
| FAB (Midgley et al., 2023) | $\mathbf{0.032 \pm 0.000}$ | $153.894 \pm 3.916$ | $0.093 \pm 0.014$ | $1204.160 \pm 147.7$ |
| GMMVI (Arenz et al., 2023) | $\mathbf{0.031 \pm 0.000}$ | $\mathbf{105.620 \pm 3.472}$ | $0.135 \pm 0.017$ | $1255.216 \pm 296.9$ |
| PIS (Zhang & Chen, 2022) | – – | – | $0.218 \pm 0.007$ | $2113.172 \pm 31.17$ |
| DDS (Vargas et al., 2023) | $0.172 \pm 0.031$ | $142.890 \pm 9.552$ | $0.131 \pm 0.001$ | $2154.884 \pm 3.861$ |
| AFT (Arbel et al., 2021) | $0.159 \pm 0.010$ | $145.138 \pm 6.061$ | $0.395 \pm 0.082$ | $2648.410 \pm 301.3$ |
| CRAFT (Arbel et al., 2021) | $0.115 \pm 0.003$ | $134.335 \pm 6.663$ | $0.257 \pm 0.024$ | $1893.926 \pm 117.3$ |
| CMCD-KL (Vargas et al., 2024) | $0.095 \pm 0.003$ | $513.339 \pm 192.4$ | – – | – – |
| NETS-AM (ours) | $0.041 \pm 0.001$ | $435.793 \pm 96.17$ | $0.0396 \pm 0.001$ | $\mathbf{407.827 \pm 69.64}$ |
| NETS-PINN (ours) | $\mathbf{0.033 \pm 0.002}$ | $388.91 \pm 141.5$ | $0.032 \pm 0.001$ | $482.393 \pm 174.6$ |
| NETS-PINN-resample (ours) | $\mathbf{0.027 \pm 0.003}$ | $343.78 \pm 65.25$ | $\mathbf{0.030 \pm 0.000}$ | $\mathbf{400.076 \pm 59.31}$ |

*Table 2.* Performance of NETS on Neal's Funnel and Mixture of Student-T distributions, measured in MMD and $\mathcal{W}_2$ distances from the true distribution. Benchmarking is in accordance with the setup of (Blessing et al., 2024). Diffusion coefficient $\epsilon_t = 5, 4$ was used for NETS-AM on the Funnel and MoS, respectively. Equivalently, $\epsilon_t = 5, 5$ were used by NETS-PINN. Bold numbers are within standard deviation the best performing. Note that NETS still has perfect sample in the $\epsilon_t \to \infty$ limit, but would require finer time discretization than the 100 sampling steps used here (see Figure 4).

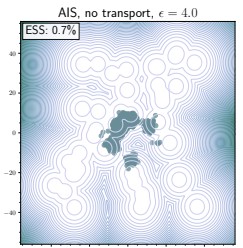
AIS, no transport, $\epsilon = 4.0$
ESS: 0.7%

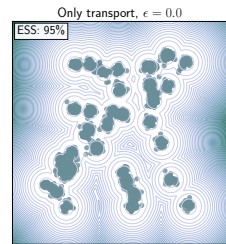
Only transport, $\epsilon = 0.0$
ESS: 95%

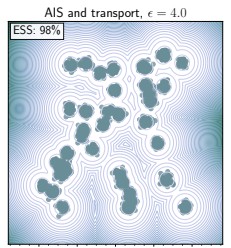
AIS and transport, $\epsilon = 4.0$
ESS: 98%

*Figure 1.* Comparison of the performance of annealed Langevin dynamics alone, transport alone, and annealed Langevin coupled with transport when sampling the 40-mode GMM from (Midgley et al., 2023). **Left**: Annealed Langevin run for 250 steps with $\epsilon_t = 4.0$, failing to capture the modes with $0\%$ ESS. **Center**: Learning using the PINN loss and sampling with 100 steps and $\epsilon_t = 0$ achieves an ESS of $95\%$. **Right**: Same learning and now sampling with $\epsilon_t = 4.0$ achieves an ESS of $98\%$.

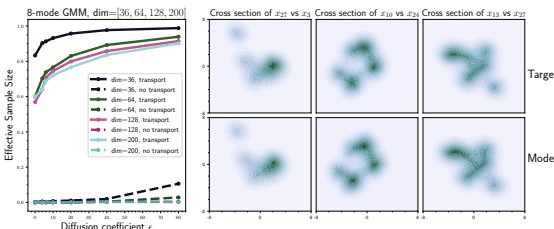

*Figure 2.* Demonstration of high-dimensional sampling with our method using the PINN loss in (25) and a study of how diffusivity impacts performance, with and without transport. **Left**: NETS can achieve high ESS through transport alone, and the effect of increased *diffusivity has more of a positive effect on performance with sampling than without*. AIS cannot achieve ESS above $\approx 0$ in high dimension. **Right**: Kernel density estimates of 2-$d$ cross sections of the high-dimensional, multimodal distribution arising from the model and ground truth.

### 3.3. Scaling on High-dimensional GMMs

In order to demonstrate that the method generalizes to high dimension, we study sampling from multimodal GMMs in higher and higher dimensions and observe how the performance scales. In addition, we are curious to understand how the factor in the sampling SDE coming from annealed Langevin dynamics, $\nabla U$, interacts with the learned drift $\hat{b}$ or $\nabla \hat{\phi}$ as we change the diffusivity. We construct 8-mode target GMMs in $d = 36, 64, 128, 200$ dimensions and learn $\hat{b}$ with the PINN loss in each scenario. We use the same feed forward neural network of width 512 and depth 4 to parameterize both $\hat{b}$ and $\hat{F}$ for all dimensions tested and train for 4000 training iterations. Figure 2 summarizes the results. On the left plot, we note that AIS on its own cannot produce any effective samples, while even in 200 dimensions, NETS works with transport alone with $60\%$ ESS. As we increase the diffusivity $\epsilon_t$ and therefore the effect of the Langevin term coming from the gradient of the potential, we note that all methods converge to nearly independent sampling, and the discrepancy in performance across dimensions is diminished. Note that the caveat to achieve this is that the step size in the SDE integrator must be taken smaller to accommodate the increased diffusivity, especially for the $\epsilon_t = 80$ data point. The number of sampling steps used to discretize the SDEs in these experiments ranged from

$K = 100$ for $\epsilon_t = 0$ up to $K = 2000$ for $\epsilon_t = 80$. Nonetheless, it suggests that diffusion can be more helpful when there is already some successful transport than without.

### 3.4. Lattice $\varphi^4$ Theory

We next apply NETS to the simulation of a statistical lattice field theory at and past the phase transition from which the lattice goes from disordered, to semi-ordered, to fully ordered (neighboring sites are highly correlated to be of the same sign and magnitude). We study the lattice $\varphi^4$ theory in $D = 2$ spacetime dimensions (Vierhaus, 2010; Albergo et al., 2019). The random variables in this circumstance are field configurations $\varphi \in \mathbb{R}^{L \times L}$, where $L$ is the extent of space and time. The interpolating energy function under which we seek to sample is defined as:

$$U_t(\varphi) = \sum_{x \sim y} |\varphi_x - \varphi_y|^2 + \sum_x \left[ m_t^2 \varphi_x^2 + \lambda_t \varphi_x^4 \right], \quad (41)$$

where the sums are taken over the lattices sites $x$, or adjacent sites $x \sim y$, and $\lambda_t$ are time-dependent parameters of the theory that define the phase of the lattice (ranging from disordered to ordered, otherwise known as magnetized). A derivation of this energy function is given in Appendix I. Importantly, sampling the lattice configurations becomes challenging when approaching the phase transition between the disordered and ordered phases. As an example, we identify the phase transition on $L = 16$ ($d = 256$) and $L = 20$ ($d = 400$) lattices and run NETS with the action matching loss, with $\hat{\phi}_t$ a simple feed forward neural network. We use the free theory $\lambda_0 = 0$ as the base distribution under which we initially draw samples. The definition of the target parameter values $m_1^2, \lambda_1$ both at the phase transition and in the ordered phase are given in the Appendix I. In Figure 3, the top row shows samples from NETS for $L = 20$ at the phase transition, where correlations begin to appear in the lattice configurations. NETS is almost 2 orders of magnitude more statistically efficient than AIS (the same setup without the transport) in sampling at the critical point, as seen in the plot showing ESS over time. Note also that NETS can produce unbiased estimates of the magnetization as compared to a Hybrid Monte Carlo (HMC) ground truth. The bottom row shows samples past the phase transition and into the ordered phase, where the lattices begin to take on

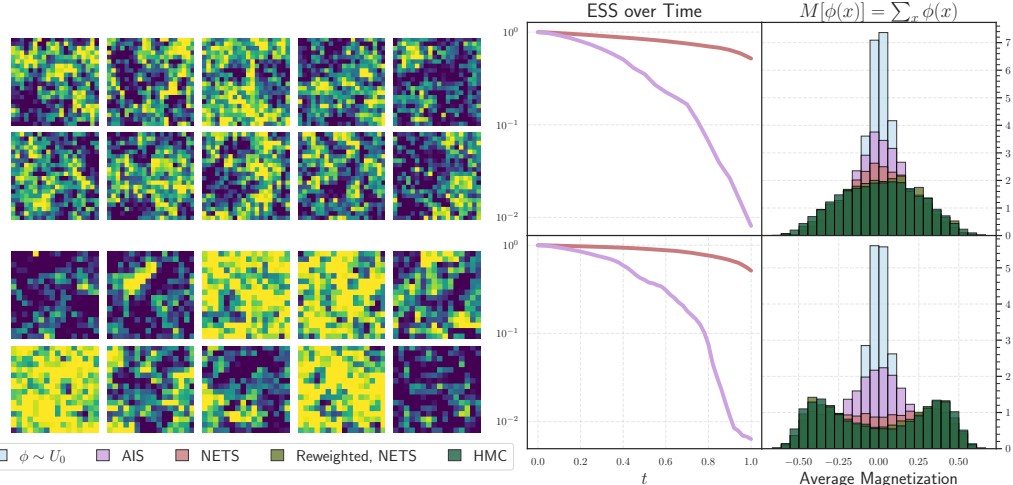

*Figure 3.* Comparison of the performance of NETS to AIS on two different settings for the study of $\varphi^4$ theory. **Top row, left**: 10 example generative lattice configurations with parameters $L = 20$, $m^2 = -1.0$, $\lambda = 0.9$, which demarcates the phase transition to the antiferromagnetic phase. **Top row, right**: Performance of AIS (purple curve) vs. NETS (red curve) in terms of effective sample size over time of integration $t$, and a histogram of the average magnetization of 4000 lattice configurations, sampled with AIS, NETS, and HMC (superposed in this order). Note that NETS is closer to the HMC target and re-weights correctly. Re-weighted AIS was not plotted because the weights were too high variance. **Bottom row**: Equivalent setup for $L = 16$, $m^2 = -1.0$, $\lambda = 0.8$, past the phase transition and into the ordered phase. Note that the field configurations generated by NETS are either all positive across lattice sites or all negative. AIS fails to sample the correct distribution, and its weights are too high variance to be used on the histogram.

either all positive or all negative values. Again in this regime, NETS is nearly 2 orders of magnitude more statistically efficient.

While NETS performs significantly better than conventional annealed samplers on the challenging field theory problem, algorithms built out of dynamical transport still experience slowdowns near phase transitions because of the difficulty of resolving the dynamics of the integrators near these critical points. As such, we need to use 1500-2000 steps in the integrator to properly resolve the dynamics of the SDE.

## Impact Statement

This paper presents work whose goal is to advance the field of Machine Learning. There are many potential societal consequences of our work, none which we feel must be specifically highlighted here.

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

# A. Proofs of Section 2

Here we provide the proofs of the statements made in Sec. 2 which, for the reader convenience, we recall.

*Proof of Proposition 2.1.* Let $f_t(x, a)$ with $(x, a) \in \mathbb{R}^{d+1}$ be the PDF of the joint process $(X_t, A_t)$ defined by the SDE (9) and (10). This PDF solves the FPE

$$\partial_t f_t = \epsilon_t \nabla \cdot (\nabla U_t f_t + \nabla f_t) + \partial_t U_t \partial_a f_t, \qquad f_{t=0}(x, a) = \delta(a)\rho_0(x). \tag{42}$$

Define

$$g_t(x) = \int_{\mathbb{R}} e^a f_t(x, a) da. \tag{43}$$

We can derive an equation for $g_t(x)$ by multiplying both sides of the FPE (42) by $e^a$ and integrating over $a \in \mathbb{R}$. Using

$$\int_{\mathbb{R}} e^a \partial_t f_t(x, a) da = \partial_t \int_{\mathbb{R}} e^a f_t(x, a) da = \partial_t g_t,$$

$$\int_{\mathbb{R}} e^a \epsilon_t \nabla \cdot (\nabla U_t f_t + \nabla f_t) da = \epsilon_t \nabla \cdot \left( \nabla U_t \int_{\mathbb{R}} e^a f_t(x, a) da + \nabla \int_{\mathbb{R}} e^a f_t(x, a) da \right)$$

$$= \epsilon_t \nabla \cdot (\nabla U_t g_t + \nabla g_t), \tag{44}$$

$$\int_{\mathbb{R}} e^a \partial_t U_t \partial_a f_t da = \partial_t U_t \int_{\mathbb{R}} e^a \partial_a f_t da$$

$$= -\partial_t U_t \int_{\mathbb{R}} e^a f_t da = -\partial_t U_t g_t,$$

where we arrived at the second equality in the third equation by integration by parts, we deduce that

$$\partial_t g_t = \epsilon_t \nabla \cdot (\nabla U_t g_t + \nabla g_t) - \partial_t U_t g_t, \qquad g_{t=0}(x) = \rho_0(x) = e^{-U_0(x) + F_0}. \tag{45}$$

The solution to this parabolic PDE is unique and it can be checked by direct substitution that it is given by

$$g_t(x) = e^{-U_t(x) + F_0}. \tag{46}$$

Note that this solution is not normalized since it contains $F_0$ rather than $F_t$. In fact it is easy to see that

$$\int_{\mathbb{R}^d} g_t(x) dx = \int_{R^{d+1}} e^a f_t(x, a) dx da = e^{-F_t + F_0}, \tag{47}$$

where the first equality follows from the definition of $g_t$ and the second from its explicit expression and the definition of the free energy that implies $\int_{\mathbb{R}^d} e^{-U_t(x)} dx = e^{-F_t}$. Equation (46) is the second equation in (11). From (46) we also deduce that, given any test function $h : \mathbb{R}^d \to \mathbb{R}$, we have

$$\frac{\int_{R^{d+1}} e^a h(x) f_t(x, a) dx da}{\int_{R^{d+1}} e^a f_t(x, a) dx da} = \frac{\int_{R^d} h(x) g_t(x) dx}{\int_{R^d} g_t(x) dx}$$

$$= \frac{\int_{R^d} h(x) e^{-U_t(x) + F_0} dx}{\int_{R^d} e^{-U_t(x) + F_0}}$$

$$= \frac{\int_{R^d} h(x) e^{-U_t(x)} dx}{\int_{R^d} e^{-U_t(x)} dx} \tag{48}$$

$$= e^{F_t} \int_{R^d} h(x) e^{-U_t(x)} dx$$

$$= \int_{R^d} h(x) \rho_t(x) dx.$$

Since by definition of $f_t(x, a)$ the left hand-side of this equation can be expressed as the ratio of expectations over $(X_t, A_t)$ in the first equation in (11) we are done. $\qquad \square$

*Proof of Proposition 2.3.* If $b_t$ satisfies (13), then $\rho_t$ satisfies the FPE (14). Since (15) is the SDE associated with this FPE, (16) holds. $\qquad\square$

*Proof of Proposition 2.4.* We can follow the same steps as in the proof of Proposition 2.1 by considering the PDF $f_t^{\hat{b}}(x, a)$ of $(X_t^{\hat{b}}, A_t^{\hat{b}})$. This PDF solves the FPE

$$\partial_t f_t^{\hat{b}} = \epsilon_t \nabla \cdot (\nabla U_t f_t^{\hat{b}} + \nabla f_t^{\hat{b}}) - \nabla \cdot (\hat{b}_t f_t^{\hat{b}}) - (\nabla \cdot \hat{b}_t - \nabla U_t - \partial_t U_t)\partial_a f_t^{\hat{b}},$$
$$f_{t=0}^{\hat{b}}(x, a) = \delta(a)\rho_0(x). \tag{49}$$

Define

$$g_t^{\hat{b}}(x) = \int_{\mathbb{R}} e^a f_t^{\hat{b}}(x, a)\, da. \tag{50}$$

We can derive an equation for $g_t^{\hat{b}}(x)$ by multiplying both sides of the FPE (49) by $e^a$ and integrating over $a \in \mathbb{R}$. Using

$$\int_{\mathbb{R}} e^a \partial_t f_t^{\hat{b}}\, da = \partial_t \int_{\mathbb{R}} e^a f_t^{\hat{b}}\, da = \partial_t g_t^{\hat{b}},$$

$$\int_{\mathbb{R}} e^a \epsilon_t \nabla \cdot (\nabla U_t f_t^{\hat{b}} + \nabla f_t^{\hat{b}})\, da = \epsilon_t \nabla \cdot \left( \nabla U_t \int_{\mathbb{R}} e^a f_t^{\hat{b}}\, da + \nabla \int_{\mathbb{R}} e^a f_t^{\hat{b}}\, da \right),$$

$$-\int_{\mathbb{R}} e^a \nabla \cdot (\hat{b}_t f_t^{\hat{b}})\, da = -\nabla \cdot \left( \hat{b}_t \int_{\mathbb{R}} e^a f_t^{\hat{b}}\, da \right)$$
$$= -\nabla \cdot (\hat{b}_t g_t^{\hat{b}}), \tag{51}$$

$$-\int_{\mathbb{R}} e^a (\nabla \cdot \hat{b}_t - \nabla U_t - \partial_t U_t)\partial_a f_t^{\hat{b}}\, da = -(\nabla \cdot \hat{b}_t - \nabla U_t - \partial_t U_t) \int_{\mathbb{R}} e^a \partial_a f_t^{\hat{b}}\, da$$
$$= (\nabla \cdot \hat{b}_t - \nabla U_t - \partial_t U_t) \int_{\mathbb{R}} e^a f_t^{\hat{b}}\, da$$
$$= (\nabla \cdot \hat{b}_t - \nabla U_t - \partial_t U_t) g_t^{\hat{b}},$$

where we arrived at the second equality in the fourth equation by integration by parts, we deduce that

$$\partial_t g_t^{\hat{b}} = \epsilon_t \nabla \cdot (\nabla U_t g_t^{\hat{b}} + \nabla g_t^{\hat{b}}) - \nabla \cdot (\hat{b}_t g_t^{\hat{b}}) + (\nabla \cdot \hat{b}_t - \nabla U_t - \partial_t U_t)g_t^{\hat{b}},$$
$$g_{t=0}^{\hat{b}}(x) = \rho_0(x) = e^{-U_0(x)+F_0}. \tag{52}$$

The solution to this parabolic PDE is unique and it can be checked by direct substitution that it is given by

$$g_t^{\hat{b}}(x) = e^{-U_t(x)+F_0}. \tag{53}$$

This solution is not normalized since it contains $F_0$ rather than $F_t$, and it is easy to see that

$$\int_{\mathbb{R}^d} g_t^{\hat{b}}(x)\, dx = \int_{R^{d+1}} e^a f_t^{\hat{b}}(x, a)\, dx\, da = e^{-F_t+F_0}. \tag{54}$$

where the first equality follows from the definition of $g_t^{\hat{b}}$ and the second from its explicit expression and the definition of the free energy that implies $\int_{\mathbb{R}^d} e^{-U_t(x)}\, dx = e^{-F_t}$. Equation (54) is the second equation in (21). From (53) we also deduce

that, given any test function $h : \mathbb{R}^d \to \mathbb{R}$, we have

$$
\begin{aligned}
\frac{\int_{\mathbb{R}^{d+1}} e^a h(x) f_t^{\hat{b}}(x,a) dx da}{\int_{\mathbb{R}^{d+1}} e^a f_t^{\hat{b}}(x,a) dx da} &= \frac{\int_{\mathbb{R}^d} h(x) g_t^{\hat{b}}(x) dx}{\int_{\mathbb{R}^d} g_t^{\hat{b}}(x) dx} \\
&= \frac{\int_{\mathbb{R}^d} h(x) e^{-U_t(x)+F_0} dx}{\int_{\mathbb{R}^d} e^{-U_t(x)+F_0}} \\
&= \frac{\int_{\mathbb{R}^d} h(x) e^{-U_t(x)} dx}{\int_{\mathbb{R}^d} e^{-U_t(x)} dx} \\
&= e^{F_t} \int_{\mathbb{R}^d} h(x) e^{-U_t(x)} dx \\
&= \int_{\mathbb{R}^d} h(x) \rho_t(x) dx.
\end{aligned}
\tag{55}
$$

Since by definition of $f_t^{\hat{b}}(x,a)$ the left hand-side of this equation can be expressed as the ratio of expectations over $(X_t^{\hat{b}}, A_t^{\hat{b}})$ in the first equation in (21) we are done. $\qquad\square$

*Proof of Proposition 2.5.* Clearly the minimum value of (25) is zero and the minimizing pair $(\hat{b}, \hat{F})$ must satisfy

$$
\nabla \cdot \hat{b}_t - \nabla U_t \cdot \hat{b}_t - \partial_t U_t + \partial_t \hat{F}_t = 0
\tag{56}
$$

By multiplying both sides of this equation by $\rho_t$ is can be written as

$$
\nabla \cdot (\hat{b}_t \rho_t) - \partial_t U_t \rho_t + \partial_t \hat{F}_t \rho_t = 0
\tag{57}
$$

This equation requires a solvability condition obtained by integrating it over $\mathbb{R}^d$. This gives

$$
-\int_{\mathbb{R}^d} \partial_t U_t(x) \rho_t(x) dx + \partial_t \hat{F}_t = 0,
\tag{58}
$$

which, by (3), implies that $\partial_t \hat{F}_t = \partial_t F_t$. In turn, this implies that (57) is equivalent to (13), i.e. $\hat{b}_t$ solves (13). $\qquad\square$

*Proof of Proposition 2.6.* Consider

$$
D_{\mathrm{KL}}(\hat{\rho}_t || \rho_t) = \int_{\mathbb{R}^d} \log\left(\frac{\hat{\rho}_t(x)}{\rho_t(x)}\right) \hat{\rho}_t(x) dx
\tag{59}
$$

where $\hat{\rho}_t$ satisfies (27). Taking the time-derivative of this expression we deduce that (using (27), $\rho_t(x) = e^{-U_t(x)+F_t}$, the

identity $\epsilon_t \nabla \cdot (\nabla U_t \hat{\rho}_t + \nabla \hat{\rho}_t) = \epsilon_t \nabla \cdot (\rho_t \nabla(\hat{\rho}_t/\rho_t))$, and multiple integrations by parts)

$$\partial_t D_{\mathrm{KL}}(\hat{\rho}_t || \rho_t) = \int_{\mathbb{R}^d} \left[ \log\left(\frac{\hat{\rho}_t(x)}{\rho_t(x)}\right) \partial_t \hat{\rho}_t(x) - \frac{\partial_t \rho_t(x)}{\rho_t(x)} \hat{\rho}_t(x) \right] dx$$

$$= \int_{\mathbb{R}^d} \log\left(\frac{\hat{\rho}_t(x)}{\rho_t(x)}\right) \cdot \left( -\nabla \cdot (\hat{b}_t(x)\hat{\rho}_t(x)) + \epsilon_t \nabla \cdot (\nabla U_t(x)\hat{\rho}_t(x) + \nabla \hat{\rho}_t(x)) \right) dx$$

$$+ \int_{\mathbb{R}^d} (\partial_t U_t(x) - \partial_t F_t) \hat{\rho}_t(x) dx$$

$$= \int_{\mathbb{R}^d} \left[ \hat{b}_t(x) \cdot \nabla \log\left(\frac{\hat{\rho}_t(x)}{\rho_t(x)}\right) + \partial_t U_t - \partial_t F_t \right] \hat{\rho}_t(x) dx$$

$$- \epsilon_t \int_{\mathbb{R}^d} \nabla \log\left(\frac{\hat{\rho}_t(x)}{\rho_t(x)}\right) \cdot \nabla \left(\frac{\hat{\rho}_t(x)}{\rho_t(x)}\right) \rho_t(x) dx \tag{60}$$

$$= \int_{\mathbb{R}^d} \left[ \hat{b}_t(x) \cdot \nabla \hat{\rho}_t(x) + \left( \hat{b}_t(x) \cdot \nabla U_t(x) + \partial_t U_t - \partial_t F_t \right) \hat{\rho}_t(x) \right] dx$$

$$- \epsilon_t \int_{\mathbb{R}^d} \left| \nabla \log\left(\frac{\hat{\rho}_t(x)}{\rho_t(x)}\right) \right|^2 \hat{\rho}_t(x) dx$$

$$= \int_{\mathbb{R}^d} \left[ -\nabla \cdot \hat{b}_t(x) + \hat{b}_t(x) \cdot \nabla U_t(x) + \partial_t U_t - \partial_t F_t \right] \hat{\rho}_t(x) dx$$

$$- \epsilon_t \int_{\mathbb{R}^d} \left| \nabla \log\left(\frac{\hat{\rho}_t(x)}{\rho_t(x)}\right) \right|^2 \hat{\rho}_t(x) dx.$$

Therefore

$$D_{\mathrm{KL}}(\hat{\rho}_{t=1} || \rho_1) = \int_0^1 \int_{\mathbb{R}^d} \left[ -\nabla \cdot \hat{b}_t(x) + \hat{b}_t(x) \cdot \nabla U_t(x) + \partial_t U_t - \partial_t F_t \right] \hat{\rho}_t(x) dx dt$$

$$- \int_0^1 \epsilon_t \int_{\mathbb{R}^d} \left| \nabla \log\left(\frac{\hat{\rho}_t(x)}{\rho_t(x)}\right) \right|^2 \hat{\rho}_t(x) dx dt$$

$$\leq \int_0^1 \int_{\mathbb{R}^d} \left[ -\nabla \cdot \hat{b}_t(x) + \hat{b}_t(x) \cdot \nabla U_t(x) + \partial_t U_t - \partial_t F_t \right] \hat{\rho}_t(x) dx dt \tag{61}$$

$$\leq \left[ \int_0^1 \int_{\mathbb{R}^d} \left| -\nabla \cdot \hat{b}_t(x) + \hat{b}_t(x) \cdot \nabla U_t(x) + \partial_t U_t - \partial_t F_t \right|^2 \hat{\rho}_t(x) dx dt \right]^{1/2}$$

$$= \sqrt{L_{\mathrm{PINN}}^{T=1}(\hat{b}, F)}$$

which gives (28). To establish (29) observe that

$$L_{\mathrm{PINN}}^1(\hat{b}, F)$$

$$= \int_0^1 \int_{\mathbb{R}^d} \left| \nabla \cdot \hat{b}_t(x) - \hat{b}_t(x) \cdot \nabla U_t(x) - \partial_t U_t + \partial_t F_t \right|^2 \hat{\rho}_t(x) dx dt$$

$$\leq 2 \int_0^1 \int_{\mathbb{R}^d} \left[ \left| \nabla \cdot \hat{b}_t(x) - \hat{b}_t(x) \cdot \nabla U_t(x) - \partial_t U_t + \partial_t \hat{F}_t \right|^2 + \left| \partial_t F_t - \partial_t \hat{F}_t \right|^2 \right] \hat{\rho}_t(x) dx dt \tag{62}$$

$$= 2 L_{\mathrm{PINN}}^1(\hat{b}, \hat{F}) + 2 \int_0^1 |\partial_t F_t - \partial_t \hat{F}_t|^2 dt$$

Therefore, if $\int_0^1 |\partial_t \hat{F}_t - \partial_t F_t|^2 dt \leq \delta$, we have

$$L_{\mathrm{PINN}}^1(\hat{b}, F) \leq 2 L_{\mathrm{PINN}}^1(\hat{b}, \hat{F}) + 2\delta \tag{63}$$

Combining this bound with (28) gives (29). $\quad\square$

## B. Time-discretized Version of Proposition 2.4

Here we show how to generalize the result in Proposition 2.4 if we time discretize the SDE in (19) using Euler-Marayuma scheme and use some suitable time-discretized version of the ODE (20).

**Proposition B.1.** *Let $0 = t_0 < t_1 < \cdots < t_K = 1$ be a time grid on $[0,1]$, denote $\Delta t_k = t_{k+1} - t_k$ for $k = 0, \ldots, K-1$, set $\tilde{X}_0^{\hat{b}} \sim \rho_0$ and $\tilde{A}_0^{\hat{b}} = 0$, and for $k = 0, \ldots, K-1$ define $\tilde{X}_{t_{k+1}}^{\hat{b}}, \tilde{A}_{t_{k+1}}^{\hat{b}}$ recursively via*

$$\tilde{X}_{t_{k+1}}^{\hat{b}} = \tilde{X}_{t_k}^{\hat{b}} - \epsilon_{t_k} \nabla U_{t_k}(\tilde{X}_{t_k}^{\hat{b}}) \Delta t_k + \hat{b}_{t_k}(\tilde{X}_{t_k}^{\hat{b}}) \Delta t_k + \sqrt{2\epsilon_{t_k}}(W_{t_{k+1}} - W_{t_k}), \tag{64}$$

$$\tilde{A}_{t_{k+1}}^{\hat{b}} = \tilde{A}_{t_k}^{\hat{b}} + U_{t_k}(\hat{X}_{t_k}^{\hat{b}}) - U_{t_{k+1}}(\hat{X}_{t_{k+1}}^{\hat{b}}) + R_k^+(\tilde{X}_{t_k}^{\hat{b}}, \tilde{X}_{t_{k+1}}^{\hat{b}}) - R_k^-(\tilde{X}_{t_{k+1}}^{\hat{b}}, \tilde{X}_{t_k}^{\hat{b}}), \tag{65}$$

*where we defined*

$$R_k^{\pm}(x, y) = \frac{1}{4\epsilon_{t_k} \Delta t_k} \left| y - x + \Delta t_k(\epsilon_{t_k} \nabla U_{t_k}(x) \mp b_{t_k}(x)) \right|^2 \tag{66}$$

*Then for all $k = 0, \ldots, K$ and any test function $h : \mathbb{R}^d \to \mathbb{R}$, we have*

$$\int_{\mathbb{R}^d} h(x) \rho_{t_k}(x) dx = \frac{\mathbb{E}[e^{\tilde{A}_{t_k}^{\hat{b}}} h(\tilde{X}_{t_k}^{\hat{b}})]}{\mathbb{E}[e^{\tilde{A}_{t_k}^{\hat{b}}}]}, \qquad Z_{t_k} = e^{-F_{t_k}} = \mathbb{E}[e^{\tilde{A}_{t_k}^{\hat{b}}}], \tag{67}$$

*where the expectations are taken over the law of $(\tilde{X}_{t_k}^{\hat{b}}, \tilde{A}_{t_k}^{\hat{b}})$*

Note that the weights in (67) correct for the bias coming for both the time evolution of $U_t(x)$ and the fact that the Euler-Maruyama update in (64) does not satisfy the detailed-balance condition locally. It cab be checked by direct calculation that (65) is a consistent time-discretization of the ODE (20).

*Proof.* For simplicity of notations we will prove (67) for $k = K$: the argument for all the other $k = 1, \ldots, K-1$ is similar. The update rule in (65) implies that

$$\tilde{A}_{t_K}^{\hat{b}} = \sum_{k=0}^{K-1} \left( U_{t_k}(\hat{X}_{t_k}^{\hat{b}}) - U_{t_{k+1}}(\hat{X}_{t_{k+1}}^{\hat{b}}) + R_k^+(\tilde{X}_{t_k}^{\hat{b}}, \tilde{X}_{t_{k+1}}^{\hat{b}}) - R_k^-(\tilde{X}_{t_{k+1}}^{\hat{b}}, \tilde{X}_{t_k}^{\hat{b}}) \right)$$

$$= U_0(\tilde{X}_{t_0}^{\hat{b}}) - U_K(\tilde{X}_{t_K}^{\hat{b}}) + \sum_{k=0}^{K-1} \left( R_k^+(\tilde{X}_{t_k}^{\hat{b}}, \tilde{X}_{t_{k+1}}^{\hat{b}}) - R_k^-(\tilde{X}_{t_{k+1}}^{\hat{b}}, \tilde{X}_{t_k}^{\hat{b}}) \right), \tag{68}$$

Now, given the test function $h : \mathbb{R}^d \to \mathbb{R}$, consider

$$I[h] \equiv \mathbb{E}\left[ e^{\tilde{A}_{t_K}^{\hat{b}}} h(\tilde{X}_{t_K}^{\hat{b}}) \right] \tag{69}$$

Since the transition probability density function of the Euler-Maruyama update in (64) reads

$$\rho_{t_k}^+(x_{k+1}|x_k) = (4\pi\epsilon_{t_k} \Delta t_k)^{-d/2} \exp\left( -R_k^+(x_k, x_{k+1}) \right), \tag{70}$$

the joint probability density function of the path $(\tilde{X}_{t_0}^{\hat{b}}, \tilde{X}_{t_1}^{\hat{b}}, \ldots, \tilde{X}_{t_K}^{\hat{b}})$ is given by

$$\rho(x_0, \ldots, x_K) = \exp\left( -U_0(x_0) + F_0 \right) \prod_{k=0}^{K-1} \rho_{t_k, \Delta t_k}^+(x_{k+1}|x_k)$$

$$= C \exp\left( -U_0(x_0) + F_0 - \sum_{k=0}^{K-1} R_k^+(x_k, x_{k+1}) \right) \tag{71}$$

where $C = \prod_{k=0}^{K-1} (4\pi\epsilon_{t_k} \Delta t_k)^{-d/2}$. We can use this density along with the explicit expression for $\tilde{A}_{T_K}^{\hat{b}}$ in (68) to express the expectation (69) as an integral over $\rho(x_0, x_1, \ldots, x_K)$

$$I[h] = C \int_{\mathbb{R}^{d(K+1)}} dx_0 \cdots dx_K \exp\left( -U_0(x_0) + F_0 - \sum_{k=0}^{K-1} R_k^+(x_k, x_{k+1}) \right)$$

$$\times \exp\left( U_0(x_0) - U_K(x_K) + \sum_{k=0}^{K-1} \left( R_k^+(x_k, x_{k+1}) - R_k^-(x_{k+1}, x_k) \right) \right) h(x_K) \tag{72}$$

where the second exponential comes from the factor $e^{\tilde{A}^{\hat{b}}_K}$. (72) simplifies into

$$I[h] = C \int_{\mathbb{R}^{d(K+1)}} dx_0 \cdots dx_K \exp\left(-U_K(x_K) + F_0 - \sum_{k=0}^{K-1} R_k^-(x_{k+1}, x_k)\right) h(x_K) \tag{73}$$

In this expression we recognize a product of factors involving

$$\rho_{t_k}^-(x_k | x_{k+1}) = (4\pi\epsilon_{t_k})^{-d/2} \exp\left(-R_k^-(x_{k+1}, x_k)\right), \tag{74}$$

which is the transition probability density function of the time-reversed update

$$\tilde{Y}_{t_k}^{\hat{b}} = \tilde{Y}_{t_{k+1}}^{\hat{b}} - \epsilon_{t_k} \nabla U_{t_k}(\tilde{Y}_{t_{k+1}}^{\hat{b}})\Delta t_k - \hat{b}_{t_k}(\tilde{Y}_{t_{k+1}}^{\hat{b}})\Delta t_k + \sqrt{2\epsilon_{t_k}}(W_{t_{k+1}} - W_{t_k}). \tag{75}$$

This implies in particular that we can perform the integrals in (73) sequentially over $x_0, x_1, .., x_{K-1}$ to be left with

$$I[h] = \int_{\mathbb{R}^d} \exp\left(-U_K(x_K) + F_0\right) h(x_K) dx_K \tag{76}$$

Therefore

$$I[1] = \int_{\mathbb{R}^d} \exp\left(-U_K(x_K) + F_0\right) dx_K = e^{-F_K + F_0}, \tag{77}$$

which is the second equation in (67), and

$$\frac{I[h]}{I[1]} = e^{F_K - F_0} \int_{\mathbb{R}^d} \exp\left(-U_K(x_K) + F_0\right) h(x_K) dx_K = \int_{\mathbb{R}^d} h(x)\rho_{t_K}(x) dx \tag{78}$$

which is the first equation in (67). $\qquad\square$

## C. Solving for the Optimal Drift via Feynman-Kac Formula

Without loss of generality, we can always look for a solution to (23) in the form of $b_t(x) = \nabla\phi_t(x)$, so that this equation becomes the Poisson equation

$$\Delta\phi_t - \nabla U_t \cdot \nabla\phi_t = \partial_t U_t - \partial_t F_t. \tag{79}$$

The solution to this equation can be expressed via Feynman-Kac formula:

**Proposition C.1.** *Let $X_\tau^{t,x}$ satisfy the following SDE*

$$dX_\tau^{t,x} = -\nabla U_t(X_\tau^{t,x})d\tau + \sqrt{2}dW_\tau, \qquad X_{\tau=0}^{t,x} = x \tag{80}$$

*where $U_t$ is evaluated fixed at $t \in [0,1]$ fixed. Assume geometric ergodicity of the semi-group associated with (80), i.e. the probability distribution of the solutions to this SDE converges exponentially fast towards their unique equilibrium distribution with density $\rho_t(x)$. Then for all $(t, x) \in [0,1] \times \mathbb{R}^d$ we have*

$$\phi_t(x) = \int_0^\infty \mathbb{E}\left[\partial_t F_t - \partial_t U_t(X_\tau^{t,x})\right] d\tau \tag{81}$$

*where the expectation is taken over the law of $X_\tau^{t,x}$.*

*Proof.* By Ito formula,

$$\begin{aligned}
d\phi_t(X_\tau^{t,x}) &= \left(\Delta\phi_t(X_\tau^{t,x}) - \nabla U_t(X_\tau^{t,x}) \cdot \nabla\phi_t(X_\tau^{t,x})\right)d\tau + \sqrt{2}\nabla\phi_t(X_\tau^{t,x}) \cdot dW_\tau \\
&= \left(\partial_t U_t(X_\tau^{t,x}) - \partial_t F_t\right) d\tau + \sqrt{2}\nabla\phi_t(X_\tau^{t,x}) \cdot dW_\tau
\end{aligned} \tag{82}$$

where the differential is taken with respect to $\tau$ at $t$ fixed, and we used (79) to get the second equality. If we integrate this relation on $\tau \in [0, T]$ and take expectation, we deduce that

$$\mathbb{E}\left[\phi_t(X_T^{t,x})\right] - \phi_t(x) = \int_0^T \mathbb{E}\left[\partial_t U_t(X_\tau^{t,x}) - \partial_t F_t\right] d\tau \tag{83}$$

where we use Ito isometry to zero the expectation of the martingale term involving $\sqrt{2}\nabla\phi_t(X_\tau^{t,x}) \cdot dW_\tau$. If we let $T \to \infty$, by ergodicty the first term at the left hand side converges towards a constant independent of $(t, x)$ which we can neglect – this fixes the gauge of the solution to (79) which is unique only up to a constant. What remains in this limit is the expression (81). Note that the integral in this expression converges since $\mathbb{E}\left[\partial_t U_t(X_\tau^{t,x})\right] \to \partial_t F_t$ exponentially fast as $\tau \to \infty$ by assumption of geometric ergodicity. $\qquad\square$

**Example: moving Gaussian distribution.** Let us consider the case where

$$U_t(x) = \tfrac{1}{2}(x - b_t)^T A_t(x - b_t),\tag{84}$$

where $b_t \in \mathbb{R}^d$ is a time-dependent vector field and $A_t = A_t^T \in \mathbb{R}^d \times \mathbb{R}^d$ is a time-dependent positive-definite matrix: we assume that both $b_t$ and $A_t$ are $C^1$ in time, and also that $\dot{A}_t A_t = A_t \dot{A}_t$. The free energy in this example is

$$F_t = -\log Z_t, \qquad Z_t = (2\pi)^{d/2} |\det A_t|^{-1/2},\tag{85}$$

so that

$$\partial_t U_t(x) = -\dot{b}_t^T A_t(x - b_t) + \tfrac{1}{2}(x - b_t)^T \dot{A}_t(x - b_t), \qquad \partial_t F_t = \tfrac{1}{2}\mathrm{tr}(A_t^{-1}\dot{A}_t).\tag{86}$$

In this case, the SDE (80) reads

$$dX_\tau^{t,x} = -A_t(X_\tau^{t,x} - b_t)d\tau + \sqrt{2}dW_\tau, \qquad X_{\tau=0}^{t,x} = x,\tag{87}$$

and its solution is

$$X_\tau^{t,x} = e^{-A_t\tau}x + (1 - e^{-A_t\tau})b_t + \sqrt{2}\int_0^\tau e^{-A_t(\tau-\tau')}dW_{\tau'}.\tag{88}$$

This implies that (using Ito isometry)

$$\begin{aligned}
\mathbb{E}\left[\partial_t U_t(X_\tau^{t,x})\right] &= -\dot{b}_t^T A_t e^{-A_t\tau}(x - b_t) + \tfrac{1}{2}(x - b_t)^T e^{-A_t\tau}\dot{A}_t e^{-A_t\tau}(x - b_t) \\
&\quad + \int_0^\tau \mathrm{tr}\left(e^{-A_t\tau}\dot{A}_t e^{-A_t\tau}\right)d\tau \\
&= -\dot{b}_t^T A_t e^{-A_t\tau}(x - b_t) + \tfrac{1}{2}(x - b_t)^T e^{-A_t\tau}\dot{A}_t e^{-A_t\tau}(x - b_t) \\
&\quad + \tfrac{1}{2}\mathrm{tr}\left(A_t^{-1}\dot{A}_t\right) - \tfrac{1}{2}\mathrm{tr}(A_t^{-1}\dot{A}_t e^{-2A_t\tau}).
\end{aligned}\tag{89}$$

Therefore, from (81), we have (using also (85))

$$\begin{aligned}
\phi_t(x) &= \int_0^\infty \Big(\dot{b}_t^T A_t e^{-A_t\tau}(x - b_t) - \tfrac{1}{2}(x - b_t)^T e^{-A_t\tau}\dot{A}_t e^{-A_t\tau}(x - b_t) \\
&\qquad\qquad + \tfrac{1}{2}\mathrm{tr}(A_t^{-1}\dot{A}_t e^{-2A_t\tau})\Big)d\tau \\
&= \dot{b}_t \cdot (x - b_t) - \tfrac{1}{4}(x - b_t)^T \dot{A}_t A_t^{-1}(x - b_t) + \tfrac{1}{4}\mathrm{tr}(A_t^{-1}\dot{A}_t A_t^{-1}).
\end{aligned}\tag{90}$$

This solution checks out since it implies that

$$-\nabla U_t(x) \cdot \nabla \phi_t(x) + \Delta \phi_t(x) = -\dot{b}_t^T A_t(x - b_t) + \tfrac{1}{2}(x - b_t)^T \dot{A}_t(x - b_t) - \tfrac{1}{2}\mathrm{tr}(A_t^{-1}\dot{A}_t),\tag{91}$$

which is $\partial_t U_t(x) - \partial_t F_t$ as it should.

# D. Extensions and Generalizations

## D.1. Inertial NETS

It is straightforward to generalize Proposition D.1 so that the stochastic dynamics involves some memory/inertia:

**Proposition D.1.** *Let* $(X_t^{\hat{b},\mu}, R_t^{\hat{b},}, A_t^{\hat{b},\mu})$ *solve the coupled system of SDE/ODE*

$$dX_t^{\hat{b},\mu} = \hat{b}_t(X_t^{\hat{b},\mu})dt + R_t^{\hat{b},\mu}dt, \qquad\qquad X_0^{\hat{b},\mu} \sim \rho_0,\tag{92}$$

$$dR_t^{\hat{b},\mu} = -\mu\nabla U_t(X_t^{\hat{b},\mu})dt - \mu\epsilon_t^{-1}R_t^{\hat{b},\mu}dt + \mu\sqrt{2\epsilon_t^{-1}}dW_t, \qquad R_0^{\hat{b},\mu} \sim N(0, \mu Id),\tag{93}$$

$$dA_t^{\hat{b},\mu} = \nabla \cdot \hat{b}_t(X_t^{\hat{b},\mu})dt - \nabla U_t(X_t^{\hat{b},\mu}) \cdot \hat{b}_t(X_t^{\hat{b},\mu})dt - \partial_t U_t(X_t^{\hat{b},\mu})dt, \quad A_0^{\hat{b},\mu} = 0,\tag{94}$$

*where* $\epsilon_t > 0$ *is a time-dependent diffusion coefficient,* $\mu \geq 0$ *is a mobility coefficient, and* $W_t \in \mathbb{R}^d$ *is the Wiener process. Then for all* $t \in [0, 1]$ *and any test function* $h : \mathbb{R}^d \to \mathbb{R}$, *we have*

$$\int_{\mathbb{R}^d} h(x)\rho_t(x)dx = \frac{\mathbb{E}[e^{A_t^{\hat{b},\mu}}h(X_t^{\hat{b},\mu})]}{\mathbb{E}[e^{A_t^{\hat{b},\mu}}]}, \qquad Z_t/Z_0 = e^{-F_t+F_0} = \mathbb{E}[e^{A_t^{\hat{b},\mu}}],\tag{95}$$

*where the expectations are taken over the law of* $(X_t^{\hat{b},\mu}, A_t^{\hat{b},\mu})$.

The proof of this proposition can be found at the end of this subsection. Note that when $\hat{b} = b$, the solution to (23), (94) is simply

$$A_t^{b,\gamma} = -F_t + F_0, \tag{96}$$

i.e. the weights are again deterministic with zero variance. In general, $\hat{b}$ will not be the optimal one, in which case using the SDE in (92)-(94) gives us the extra parameter $\mu$ to play with post-training to improve the ESS. Below we show that (92)-(94) reduce to (19)-(20) in the limit as $\mu \to \infty$. It is also easy to see that, if we set $\mu = 0$ in (92)-(94), we simply get that $R_t^{\hat{b},\mu} = 0$ and hence (97) reduces to the ODE $dX_t^{\hat{b},\mu} = \hat{b}_t(X_t^{\hat{b},\mu})dt$. Finally it is worth noting that (92)-(93) can be cast into Langevin equations with some extra forces. Indeed, if we introduce the velocity $V_t^{\hat{b},\mu} = \hat{b}_t(X_t^{\hat{b},\mu}) + R_t^{\hat{b},\mu}$, (92)-(93) can be written as

$$dX_t^{\hat{b},\mu} = V_t^{\hat{b},\mu}dt \qquad\qquad\qquad\qquad X_0^{\hat{b},\mu} \sim \rho_0, \tag{97}$$

$$dV_t^{\hat{b},\mu} = -\mu\nabla U_t(X_t^{\hat{b},\mu})dt + \mu\epsilon_t^{-1}\hat{b}_t(X_t^{\hat{b},\mu})dt - \partial_t\hat{b}_t(X_t^{\hat{b},\mu})dt$$
$$\qquad + \nabla b_t(X_t^{\hat{b},\mu})V_t^{\hat{b},\mu}dt - \mu\epsilon_t^{-1}V_t^{\hat{b},\mu}dt + \mu\sqrt{2\epsilon_t^{-1}}dW_t, \quad V_0^{\hat{b},\mu} \sim N(\hat{b}_0(X_0^{\hat{b},\mu}), \mu\mathrm{Id}) \tag{98}$$

In these equations, the terms $\mu\epsilon_t^{-1}\hat{b}_t - \partial_t\hat{b}_t$ can be interpreted as non-conservative forces added to $-\mu\nabla U_t$, and the term $\nabla b_t V_t^{\hat{b},\mu}$ as an extra friction term added to $-\mu\epsilon_t^{-1}V_t^{\hat{b},\mu}$.

*Proof of Proposition D.1.* Denote by $f_t^{\hat{b},\mu}(x,r,a)$ the joint PDF of $(X_t^{\hat{b},\mu}, R_t^{\hat{b},\mu}, A_t^{\hat{b},\mu})$. This PDF satisfies the FPE

$$\partial_t f_t^{\hat{b},\mu} = -\nabla_x \cdot ([\hat{b}_t + r]f_t^{\hat{b},\mu}) + \mu\nabla U_t \cdot \nabla_r f + \mu\epsilon_t^{-1}\nabla_r \cdot (rf_t^{\hat{b},\mu} + \mu\nabla_r f_t^{\hat{b},\mu})$$
$$\qquad - (\nabla \cdot \hat{b}_t - \nabla U_t \cdot \hat{b}_t - \partial_t U_t)\partial_a f_t^{\hat{b},\mu}, \tag{99}$$
$$f_0^{\hat{b},\mu}(x,r,a) = \rho_0(x)(2\pi\mu)^{-d/2}e^{-|r|^2/(2\mu)}\delta(a).$$

Let

$$g_t^{\hat{b},\mu}(x,r) = \int_\mathbb{R} e^a f_t^{\hat{b},\mu}(x,r,a)da. \tag{100}$$

We can derive an equation for $g_t^{\hat{b},\mu}(x)$ by multiplying both sides of the FPE (99) by $e^a$ and integrating over $a \in \mathbb{R}$. Using equations similar to (51), we arrive at

$$\partial_t g_t^{\hat{b},\mu} = -\nabla_x \cdot ([\hat{b}_t + r]g_t^{\hat{b},\mu}) + \mu\nabla U_t \cdot \nabla_r f + \mu\epsilon_t^{-1}\nabla_r \cdot (rg_t^{\hat{b},\mu} + \mu\nabla_r g_t^{\hat{b},\mu})$$
$$\qquad + (\nabla \cdot \hat{b}_t - \nabla U_t \cdot \hat{b}_t - \partial_t U_t)g_t^{b,\gamma}, \tag{101}$$
$$g_0^{\hat{b},\mu}(x,r) = \rho_0(x)(2\pi\mu)^{-d/2}e^{-|r|^2/(2\mu)}.$$

Since $\rho_0(x) = e^{-U_0(x)+F_0}$, it can be checked by direct substitution that the solution to this equation is

$$g_t^{\hat{b},\mu}(x,r) = e^{-U_t(x)+F_0}(2\pi\mu)^{-d/2}e^{-|r|^2/(2\mu)}. \tag{102}$$

Therefore

$$\int_{\mathbb{R}^{2d}} g_t^{\hat{b},\mu}(x,r)dxdr = \int_{\mathbb{R}^{2d+1}} e^a f_t^{\hat{b},\mu}(x,r,a)dxdrda = e^{-F_t+F_0}, \tag{103}$$

where the first equality follows from the definition of $g_t^{\hat{b},\mu}$ and the second from its explicit expression and the definition of the free energy that implies $\int_{\mathbb{R}^d} e^{-U_t(x)}dx = e^{-F_t}$. Equation (103) is the second equation in (95). From (102) we also

deduce that, given any test function $h : \mathbb{R}^d \to \mathbb{R}$, we have

$$
\begin{aligned}
\frac{\int_{\mathbb{R}^{2d+1}} e^a h(x) f_t^{\hat{b},\mu}(x,r,a) dx dr da}{\int_{\mathbb{R}^{2d+1}} e^a f_t^{\hat{b},\mu}(x,r,a) dx dr da} &= \frac{\int_{\mathbb{R}^{2d}} h(x) g_t^{\hat{b},\mu}(x,r) dx dr}{\int_{\mathbb{R}^{2d}} g_t^{\hat{b},\mu}(x,r) dx dr} \\
&= \frac{\int_{\mathbb{R}^d} h(x) e^{-U_t(x)+F_0} dx}{\int_{\mathbb{R}^d} e^{-U_t(x)+F_0}} \\
&= \frac{\int_{\mathbb{R}^d} h(x) e^{-U_t(x)} dx}{\int_{\mathbb{R}^d} e^{-U_t(x)} dx} \\
&= e^{F_t} \int_{\mathbb{R}^d} h(x) e^{-U_t(x)} dx \\
&= \int_{\mathbb{R}^d} h(x) \rho_t(x) dx.
\end{aligned}
\tag{104}
$$

Since by definition of $f_t^{\hat{b},\mu}(x,r,a)$ the left hand-side of this equation can be expressed as the ratio of expectations over $(X_t^{\hat{b},\mu}, A_t^{\hat{b},\mu})$ in the first equation in (95) we are done. $\qquad\square$

To see what happens when $\mu \to \infty$, let us assume that $\epsilon_t = \epsilon$ (time-independent) and integrate (93) using Duhamel principle as

$$
R_t^{\hat{b},\mu} = e^{-\mu\epsilon^{-1}t} R_0^{\hat{b},\mu} - \mu \int_0^t e^{-\mu\epsilon^{-1}(t-s)} \nabla U_s(X_s^{\hat{b},\mu}) ds + \mu\sqrt{2\epsilon^{-1}} \int_0^t e^{-\mu\epsilon^{-1}(t-s)} dW_s,
\tag{105}
$$

Letting $\mu \to \infty$, we see that the first term at the right hand side of (105) tends to zero, whereas the second one gives

$$
\lim_{\mu\to\infty} \mu \int_0^t e^{-\mu\epsilon^{-1}(t-s)} \nabla U_s(X_s^{\hat{b},\mu}) ds = \epsilon \nabla U_t(X_t^{\hat{b},\mu})
\tag{106}
$$

Finally, the third term at the right hand side of (105) is a Gaussian process with covariance

$$
C_{t,t'}^\mu = 2\mu^2 \epsilon^{-1} \int_0^{\min(t,t')} e^{-\mu\epsilon^{-1}(t-s)-\mu\epsilon^{-1}(t'-s)} ds = 2\mu \left( e^{-\mu\epsilon^{-1}|t-t'|} - e^{-\mu\epsilon^{-1}(t+t')} \right)
\tag{107}
$$

As a result, given any test function $\phi_t$, we have

$$
\lim_{\mu\to\infty} \int_{[0,1]^2} \phi_t C_{t,t'}^\mu \phi_{t'} dt dt' = 2\epsilon \int_0^1 \phi_t^2 dt
\tag{108}
$$

which indicates that $C_{t,t'}^\mu$ converges weakly towards the Dirac distribution $\epsilon\delta(t-t')$. Putting these results together shows that in the limit as $\mu \to \infty$, $R_t^{\hat{b},\mu} dt$ converges weakly towards $-\epsilon\nabla U_t(X_t^{\hat{b},\mu}) dt + \sqrt{2\epsilon} dW_t$, which, if inserted in (92), reduces this equation to (19). The case where $\epsilon_t$ depends on time can be treated similarly.

### D.2. Multimarginal NETS

Let $\mathcal{U}(\alpha, x)$ be a potential depending on $\alpha \in D \subset \mathbb{R}^N$ with $N \in \mathbb{N}$ as well as $x \in \mathbb{R}^d$, and assumed to be continuously differentiable in both arguments. Assume that $e^{-\mathcal{U}(\alpha,x)}$ is integrable in $x$ for all $\alpha \in D$, and define the family of PDF

$$
\varrho(\alpha, x) = e^{-\mathcal{U}(\alpha,x)+\mathcal{F}(\alpha)}, \qquad \mathcal{F}(\alpha) = -\log \int_{\mathbb{R}^d} e^{-\mathcal{U}(\alpha,x)} dx.
\tag{109}
$$

Finally, define the family of matrix-valued $\hat{\mathcal{B}}(\alpha, x) : D \times \mathbb{R}^d \to \mathbb{R}^N \times \mathbb{R}^d$, assumed to be continuously differentiable in both arguments. These quantities allow us to give a generalization of Proposition 2.4 in which we can sample the PDF $\varrho(\alpha, x)$ along any differential path $\alpha_t \in D$:

**Proposition D.2.** *Let $\alpha : [0,1] \to D$ be a differentiable path in $D$ and define the vector field $b : [0,1] \times \mathbb{R}^d \to \mathbb{R}^d$ as*

$$\hat{b}_t^\alpha(x) = \dot{\alpha}_t^T \hat{\mathcal{B}}(\alpha_t, x) \tag{110}$$

*as well as*

$$U_t^\alpha(x) = \mathcal{U}(\alpha_t, x), \qquad F_t^\alpha = \mathcal{F}(\alpha_t), \qquad \rho_t^\alpha = \varrho(\alpha, x) = e^{-U_t^\alpha(x) + F_t^\alpha} \tag{111}$$

*Let $(X_t^{\hat{b},\alpha}, A_t^{\hat{b},\alpha})$ solve the coupled system of SDE/ODE*

$$dX_t^{\hat{b},\alpha} = \hat{b}_t^\alpha(X_t^{\hat{b},\alpha})dt - \epsilon_t \nabla U_t^\alpha(X_t^{\hat{b},\alpha})dt + \sqrt{2\epsilon_t}dW_t \qquad\qquad X_0^{\hat{b},\alpha} \sim \rho_0^\alpha, \tag{112}$$

$$dA_t^{\hat{b},\alpha} = \nabla \cdot \hat{b}_t^\alpha(X_t^{\hat{b},\alpha})dt - \nabla U_t^\alpha(X_t^{\hat{b},\alpha}) \cdot \hat{b}_t^\alpha(X_t^{\hat{b},\alpha})dt - \partial_t U_t^\alpha(X_t^{\hat{b},\alpha})dt, \quad A_0^{\hat{b},\alpha} = 0, \tag{113}$$

*where $\epsilon_t > 0$ is a time-dependent diffusion coefficient and $W_t \in \mathbb{R}^d$ is the Wiener process. Then for all $t \in [0,1]$ and any test function $h : \mathbb{R}^d \to \mathbb{R}$, we have*

$$\int_{\mathbb{R}^d} h(x)\rho_t^\alpha(x)dx = \frac{\mathbb{E}[e^{A_t^{\hat{b},\alpha}} h(X_t^{\hat{b},\alpha})]}{\mathbb{E}[e^{A_t^{\hat{b},\alpha}}]}, \qquad e^{-F_t^\alpha + F_0^\alpha} = \mathbb{E}[e^{A_t^{\hat{b},\alpha}}], \tag{114}$$

*where the expectations are taken over the law of $(X_t^{\hat{b},\alpha}, A_t^{\hat{b},\alpha})$.*

We will omit to give the proof of this proposition since it is a simple consequence of Proposition 2.4. The interest in formulating the problem in this new way is that is it easy to see that the right hand side of (113) (with $\partial_t F_t^\alpha$ added for convenience) can be written as

$$\nabla \cdot \hat{b}_t^\alpha(x) - \nabla U_t^\alpha(x) \cdot \hat{b}_t^\alpha(x) - \partial_t U_t^\alpha(x) + \partial_t F_t^\alpha$$
$$= \dot{\alpha}_t^T \left( \nabla_x \cdot \hat{\mathcal{B}}(\alpha_t, x) - \hat{\mathcal{B}}(\alpha_t, x)\nabla_x \mathcal{U}(\alpha_t, x) - \nabla_\alpha \mathcal{U}(\alpha_t, x) + \nabla_\alpha \mathcal{F}(\alpha_t) \right). \tag{115}$$

Therefore if we zero this term for all $(\alpha, x) \in D \times \mathbb{R}^d$ by picking the right $\hat{\mathcal{B}}(\alpha, x)$ we will obtain that (113) reduces to $dA_t^{\hat{b},\alpha} = -F_t^\alpha dt$, i.e. $A_t^{\hat{b},\alpha} = F_t^\alpha - F_0^\alpha$. Finding this optimal $\mathcal{B}(\alpha, x)$ can be obtained using the following result:

**Proposition D.3** (Multimarginal PINN objective). *Consider the objective for $(\hat{\mathcal{B}}, \hat{\mathcal{F}})$ given by:*

$$L_{PINN}^\alpha[\hat{\mathcal{B}}, \hat{\mathcal{F}}]$$
$$= \int_D \int_{\mathbb{R}^d} \left| \nabla_x \cdot \hat{\mathcal{B}}(\alpha, x) - \hat{\mathcal{B}}(\alpha, x)\nabla_x \mathcal{U}(\alpha, x) - \nabla_\alpha \mathcal{U}(\alpha, x) + \nabla_\alpha \hat{\mathcal{F}}(\alpha) \right|^2 \varrho(\alpha, x)f(\alpha)dxd\alpha \tag{116}$$

*where $\hat{\varrho}(\alpha, x) > 0$ is a PDF in $x$ for all $\alpha \in D$, and $f(\alpha)$ is a PDF in $\alpha$. Then $\min_{\mathcal{B},\mathcal{F}} L_{PINN}^\alpha[\hat{\mathcal{B}}, \hat{\mathcal{F}}] = 0$, and all minimizers $(\mathcal{B}, \mathcal{F})$ are such that and $\mathcal{B}(\alpha, x)$ solves*

$$\forall(\alpha, x) \in D \times \mathbb{R}^d : \quad 0 = \nabla_x \cdot \hat{\mathcal{B}}(\alpha, x) - \hat{\mathcal{B}}(\alpha, x)\nabla_x \mathcal{U}(\alpha, x) - \nabla_\alpha \mathcal{U}(\alpha, x) + \nabla_\alpha \mathcal{F}(\alpha), \tag{117}$$

*and $\mathcal{F}(\alpha)$ is the free energy (109) for all $\alpha \in D$.*

We will omit to give the proof of this proposition since it is a simple generalization of the proof of Proposition 2.5.

# E. Estimating the Drift $b_t(x) = \nabla\phi_t(x)$ via Action Matching (AM)

In general (13) is solved by many $b_t(x)$. One way to get a unique (up to a constant in space) solution to this equation is to impose that the velocity be in gradient form, i.e. set $b_t(x) = \nabla\phi_t(x)$ for some scalar-valued potential $\phi_t(x)$. If we do so, (13) can be written as $\nabla \cdot (\nabla\phi_t(x)\rho_t) = -\partial_t\rho_t$, and it is easy to see that at all times $t \in [0,1]$ the solution to this equation minimizes over $\hat{\phi}_t$ the objective

$$\int_{\mathbb{R}^d} \left[ \tfrac{1}{2}|\nabla\hat{\phi}_t(x)|^2 \rho_t(x) - \hat{\phi}_t(x)\partial_t\rho_t(x) \right]dx$$
$$= \int_{\mathbb{R}^d} \left[ \tfrac{1}{2}|\nabla\hat{\phi}_t(x)|^2 + (\partial_t U_t(x) - \partial_t F_t)\hat{\phi}_t(x) \right]\rho_t(x)dx. \tag{118}$$

If we use (3) to set $\partial_t F_t = \int_{\mathbb{R}^d} \partial_t U_t(x)\rho_t(x)dx$ we can use the objective at the right hand-side of (118) to learn $\phi_t(x)$ locally in time (or globally if we integrate this objective on $t \in [0,1]$). Alternatively, we can integrate the objective at the left hand-side of (118) over $t \in [0,T]$ and use integration by parts for the term involving $\partial_t \rho_t(x)$ to arrive at:

**Proposition E.1** (Action Matching objective). *Given any $T \in (0,1]$ consider the objective for $\hat{\phi}_t(x)$:*

$$L_{AM}^T[\hat{\phi}] = \int_0^T \int_{\mathbb{R}^d} \left[\tfrac{1}{2}|\nabla\hat{\phi}_t(x)|^2 + \partial_t\hat{\phi}_t(x)\right]\rho_t(x)dxdt \tag{119}$$
$$+ \int_{\mathbb{R}^d} \left[\hat{\phi}_0(x)\rho_0(x) - \hat{\phi}_T(x)\rho_T(x)\right]dx.$$

*Then the minimizer $\phi_t(x)$ of (25) is unique (up to a constant) and $b_t(x) = \nabla\phi_t(x)$ satisfies (13) for all $t \in [0,T]$.*

This objective is analogous to the loss presented in Neklyudov et al. (2023), but adapted to the sampling problem. In practice, we will use again $T \in (0,1]$ for annealing, but ultimately we are interested in the result at $T = 1$. Note that, unlike with the PINN objective (25), it is crucial that we use the correct $\rho_t(x)$ in the AM objective (119): that is, unlike (25), (119) cannot be turned into an off-policy objective.

*Proof.* By integrating by parts in time the term involving $\partial_t\phi_t$ in the AM objective (120), we can express is as

$$L_{AM}^T[\hat{\phi}] = \int_0^T \int_{\mathbb{R}^d} \left[\tfrac{1}{2}|\nabla\hat{\phi}_t(x)|^2\rho_t(x) - \phi_t(x)\partial_t\rho_t(x)\right]dxdt. \tag{120}$$

This is a convex objective in $\hat{\phi}$ whose minimizers satisfy

$$\nabla \cdot (\nabla\hat{\phi}_t\rho_t) = -\partial_t\rho_t. \tag{121}$$

This is (13) written in terms of $b_t(x) = \nabla\phi_t(x)$. The solution of this equation is unique up to a constant by the Fredholm alternative since its right hand-side satisfies the solvability condition $\int_{\mathbb{R}^d} \partial_t\rho_t(x)dx = 0$. $\square$

If we use $\hat{b}_t(x) = \nabla\hat{\phi}_t(x)$ in the SDEs in (19) and (20), we need to compute $\nabla \cdot \hat{b}_t(x) = \Delta\hat{\phi}_t(x)$, which is computationally costly. Fortunately, when $\epsilon_t > 0$, the calculation of this Laplacian can be avoided by using the following alternative equation for $A_t^{\hat{b}}$:

$$A_t^{\hat{b}} = \frac{1}{\epsilon_t}[\hat{\phi}_t(X_t^{\hat{b}}) - \hat{\phi}_0(X_0^{\hat{b}})] - B_t, \tag{122}$$

where

$$dB_t = \partial_t U_t(X_t^{\hat{b}})dt + \frac{1}{\epsilon_t}\partial_t\hat{\phi}_t(X_t^{\hat{b}}) + \frac{1}{\epsilon_t}|\nabla\hat{\phi}_t(X_t^{\hat{b}})|^2dt \tag{123}$$
$$+ \sqrt{\frac{2}{\epsilon_t}}\nabla\hat{\phi}_t(X_t^{\hat{b}}) \cdot dW_t.$$

To derive these equations, notice f $\hat{b}_t(x) = \nabla\hat{\phi}_t(x)$, the SDEs (19) and (20) reduce to

$$dX_t^{\hat{b}} = -\epsilon_t\nabla U_t(X_t^{\hat{b}})dt + \hat{\nabla}\phi_t(X_t^{\hat{b}})dt + \sqrt{2\epsilon_t}dW_t, \qquad \hat{X}_0^{\hat{b}} \sim \rho_0, \tag{124}$$
$$dA_t^{\hat{b}} = \Delta\hat{\phi}_t(X_t^{\hat{b}})dt - \nabla U_t(X_t^{\hat{b}}) \cdot \nabla\hat{\phi}(X_t^{\hat{b}})dt - \partial_t U_t(X_t^{\hat{b}})dt, \qquad A_0^{\hat{b}} = 0, \tag{125}$$

Since by Itô formula we have

$$d\hat{\phi}_t(X_t^{\hat{b}}) = \partial_t\hat{\phi}_t(X_t^{\hat{b}})dt - \epsilon_t\nabla\hat{\phi}_t(X_t^{\hat{b}}) \cdot \nabla U_t(X_t^{\hat{b}})dt + |\nabla\hat{\phi}_t(X_t^{\hat{b}})|^2dt \tag{126}$$
$$+ \sqrt{2\epsilon_t}\nabla\hat{\phi}_t(X_t^{\hat{b}}) \cdot dW_t + \epsilon_t\Delta\hat{\phi}_t(X_t^{\hat{b}})dt,$$

we can express

$$\Delta\hat{\phi}_t(X_t^{\hat{b}})dt = \frac{1}{\epsilon_t}d\hat{\phi}_t(X_t^{\hat{b}})dt - \frac{1}{\epsilon_t}\partial_t\hat{\phi}_t(X_t^{\hat{b}})dt + \nabla\hat{\phi}_t(X_t^{\hat{b}}) \cdot \nabla U_t(X_t^{\hat{b}})dt \tag{127}$$
$$- \frac{1}{\epsilon_t}|\nabla\hat{\phi}_t(X_t^{\hat{b}})|^2dt - \sqrt{\frac{2}{\epsilon_t}}\nabla\hat{\phi}_t(X_t^{\hat{b}}) \cdot dW_t.$$

If we insert this expression in the SDE (125), we can write it as

$$dA_t^{\hat{b}} = \frac{1}{\epsilon_t}d\hat{\phi}_t(X_t^{\hat{b}})dt + dB_t. \tag{128}$$

where $dB_t$ is given by (123). Integrating (128) gives (122).

In terms of implementation, we can write the AM loss (119) as

$$L_{\text{AM}}^T[\hat{\phi}] = \int_0^T \frac{\mathbb{E}\big[e^{A_t^{\hat{b}}}\big[\frac{1}{2}|\nabla\hat{\phi}_t(X_t^{\hat{b}})|^2 + \partial_t\phi_t(X_t^{\hat{b}})\big]\big]}{\mathbb{E}[e^{A_t^{\hat{b}}}]}dt + \frac{\mathbb{E}\big[e^{A_0^{\hat{b}}}\phi_0(X_0^{\hat{b}})\big]}{\mathbb{E}[e^{A_0^{\hat{b}}}]} - \frac{\mathbb{E}\big[e^{A_T^{\hat{b}}}\phi_T(X_T^{\hat{b}})\big]}{\mathbb{E}[e^{A_T^{\hat{b}}}]}. \tag{129}$$

These expectations can be estimated empirically over solutions to (19) and (20) with $\hat{b}_t(x) = \nabla\hat{\phi}_t(x)$. The above implementation is detailed in Algorithm 1.

## F. Link with CMCD (Vargas et al., 2024)

Consider the process $Y_t^{\hat{b}}$ solution to the SDE

$$dY_t^{\hat{b}} = \epsilon_t\nabla U_t(Y_t^{\hat{b}})dt + 2\epsilon_t\nabla\log\rho_t^{\hat{b}}(Y_t^{\hat{b}})dt + \hat{b}_t(Y_t^{\hat{b}})dt + \sqrt{2\epsilon_t}dW_t, \quad Y_0^{\hat{b}} \sim \rho_0. \tag{130}$$

where $\rho_t^{\hat{b}}$ denotes the PDF of the process $X_t^{\hat{b}}$ defined by the SDE (19), i.e. the solution to the FPE

$$\partial_t\rho_t^{\hat{b}} = \epsilon_t\nabla\cdot(\nabla U_t\rho_t^{\hat{b}} + \nabla\rho_t^{\hat{b}}) - \nabla\cdot(\hat{b}_t\rho_t^{\hat{b}}), \qquad \rho_0^{\hat{b}} = \rho_0 \tag{131}$$

The process $Y_t^{\hat{b}}$ has a simple interpretation: it is the time-reversed of the process run using the time-reversed potential $U_{1-t}$ and $-\hat{b}_{1-t}$: that is, if the additional drift $\hat{b}_t$ was the perfect one solution to (23), the law of $X^{\hat{b}} = (X_t^{\hat{b}})_{t\in[0,1]}$ and $Y^b = (Y_t^{\hat{b}})_{t\in[0,1]}$ should coincide. This suggests to learn $b$ using as objective a divergence of the path measure of $X^{\hat{b}}$ from that of $Y^{\hat{b}}$. This is essentially what is suggested in (Vargas et al., 2024), and for the reader convenience let us re-derive some of their results in our notations.

The Kullback-Leibler divergence (or relative entropy) of the path measure of $X^{\hat{b}}$ from that of $Y^{\hat{b}}$ reads

$$\text{KL}(X^{\hat{b}}\|Y^{\hat{b}}) = \frac{1}{4}\epsilon_t\int_0^1 \mathbb{E}\big[|\nabla U_t(X_t^{\hat{b}}) + \nabla\log\rho_t^{\hat{b}}(X_t^{\hat{b}})|^2\big]dt \tag{132}$$

This objective is akin to the one used in score-based diffusion modeling (SBDM) and simply says that one way to adjust $\hat{b}$ is by matching the score of $\rho_t^{\hat{b}}$ to that of $\rho_t$. As written (132) is not explicit since we do not know $\nabla\log\rho_t^{\hat{b}}$. We can however make it explicit after a few manipulations similar to those used in SBDM. To this end, notice first that, by Ito formula, we have

$$d\log\rho_t^{\hat{b}}(X_t^{\hat{b}}) = \nabla\log\rho_t^{\hat{b}}(X_t^{\hat{b}})\cdot(-\epsilon_t\nabla U_t(X_t^{\hat{b}}) + \hat{b}_t(X_t^{\hat{b}}))dt + \epsilon_t\Delta\log\rho_t^{\hat{b}}(X_t^{\hat{b}})dt \\ + \sqrt{2\epsilon_t}\nabla\log\rho_t^{\hat{b}}(X_t^{\hat{b}})\cdot dW_t \tag{133}$$

which implies that

$$\epsilon_t\nabla\log\rho_t^{\hat{b}}(X_t^{\hat{b}})\cdot\nabla U_t((X_t^{\hat{b}})dt = -d\log\rho_t^{\hat{b}}(X_t^{\hat{b}}) + \nabla\log\rho_t^{\hat{b}}(X_t^{\hat{b}})\cdot\hat{b}_t(X_t^{\hat{b}})dt \\ + \epsilon_t\Delta\log\rho_t^{\hat{b}}(X_t^{\hat{b}})dt + \sqrt{2\epsilon_t}\nabla\log\rho_t^{\hat{b}}(X_t^{\hat{b}})\cdot dW_t \tag{134}$$

Inserting this expression in (132) after expanding the square, and noticing that the martingale term involving $dW_t$ disappears by Ito isometry and that the term $d\log\rho_t^{\hat{b}}(X_t^{\hat{b}})$ can be integrated in time we arrive at

$$\begin{aligned} \text{KL}(X^{\hat{b}}\|Y^{\hat{b}}) &= \frac{1}{4}\epsilon_t\int_0^1 \mathbb{E}\big[|\nabla U_t(X_t^{\hat{b}})|^2 + |\nabla\log\rho_t^{\hat{b}}(X_t^{\hat{b}})|^2 + 2\nabla U_t(X_t^{\hat{b}})\cdot\nabla\log\rho_t^{\hat{b}}(X_t^{\hat{b}})\big]dt \\ &= \frac{1}{4}\int_0^1 \mathbb{E}\big[\epsilon_t|\nabla U_t(X_t^{\hat{b}})|^2 + \epsilon_t|\nabla\log\rho_t^{\hat{b}}(X_t^{\hat{b}})|^2 + \epsilon_t\nabla U_t(X_t^{\hat{b}})\cdot\nabla\log\rho_t^{\hat{b}}(X_t^{\hat{b}})\big]dt \\ &\quad + \frac{1}{4}\int_0^1 \mathbb{E}\big[\nabla\log\rho_t^{\hat{b}}(X_t^{\hat{b}})\cdot\hat{b}_t(X_t^{\hat{b}}) + \epsilon_t\Delta\log\rho_t^{\hat{b}}(X_t^{\hat{b}})\big]dt \\ &\quad + \frac{1}{4}\mathbb{E}[\log\rho_0(X_0)] - \frac{1}{4}\mathbb{E}[\log\rho_1^{\hat{b}}(X_1)] \end{aligned} \tag{135}$$

where we used $\rho_0^{\hat{b}} = \rho_0$. We can now use the following identities, each obtained using $\rho_t^{\hat{b}} \nabla \log \rho_t^{\hat{b}} = \nabla \rho_t^{\hat{b}}$ and one integration by parts:

$$
\begin{aligned}
\mathbb{E}\big[|\nabla \log \rho_t^{\hat{b}}(X_t^{\hat{b}})|^2\big] &= \int_{\mathbb{R}^d} |\nabla \log \rho_t^{\hat{b}}(x)|^2 \rho_t^{\hat{b}}(x) dx \\
&= \int_{\mathbb{R}^d} \nabla \log \rho_t^{\hat{b}}(x) \cdot \nabla \rho_t^{\hat{b}}(x) dx \\
&= -\int_{\mathbb{R}^d} \Delta \log \rho_t^{\hat{b}}(x) \rho_t^{\hat{b}}(x) dx \\
&= -\mathbb{E}\big[\Delta \log \rho_t^{\hat{b}}(X_t^{\hat{b}})\big],
\end{aligned}
\tag{136}
$$

$$
\begin{aligned}
\mathbb{E}\big[\nabla U_t(X_t^{\hat{b}}) \cdot \nabla \log \rho_t^{\hat{b}}(X_t^{\hat{b}})\big] &= \int_{\mathbb{R}^d} \nabla U_t(x) \cdot \nabla \log \rho_t^{\hat{b}}(x) \rho_t^{\hat{b}}(x) dx \\
&= \int_{\mathbb{R}^d} \nabla U_t(x) \cdot \nabla \rho_t^{\hat{b}}(x) dx \\
&= -\int_{\mathbb{R}^d} \Delta U_t(x) \rho_t^{\hat{b}}(x) dx \\
&= -\mathbb{E}\big[\Delta U_t(X_t^{\hat{b}})\big]
\end{aligned}
\tag{137}
$$

and

$$
\begin{aligned}
\mathbb{E}\big[\nabla \log \rho_t^{\hat{b}}(X_t^{\hat{b}}) \cdot \hat{b}_t(X_t^{\hat{b}})\big] &= \int_{\mathbb{R}^d} \nabla \log \rho_t^{\hat{b}}(x) \cdot \hat{b}_t(x) \rho_t^{\hat{b}}(x) dx \\
&= -\int_{\mathbb{R}^d} \nabla \cdot \hat{b}_t(x) \rho_t^{\hat{b}}(x) dx \\
&= -\mathbb{E}\big[\nabla \cdot \hat{b}_t(X_t^{\hat{b}})\big]
\end{aligned}
\tag{138}
$$

Inserting these expressions in (135), it reduces to

$$
\begin{aligned}
\mathrm{KL}(X^{\hat{b}} \| Y^{\hat{b}}) &= \frac{1}{4} \int_0^1 \mathbb{E}\big[\epsilon_t |\nabla U_t(X_t^{\hat{b}})|^2 - \epsilon_t \Delta U_t(X_t^{\hat{b}}) - \nabla \cdot b_t(X_t^{\hat{b}})\big] dt \\
&\quad + \frac{1}{4} \mathbb{E}[\log \rho_0(X_0)] - \frac{1}{4} \mathbb{E}[\log \rho_1^{\hat{b}}(X_1)]
\end{aligned}
\tag{139}
$$

This objective is still not practical because it involves $\log \rho_1^{\hat{b}}$, which is unknown. There is however a simple way to fix this, by adding a term in the Kullback-Leibler divergence (132)

$$
\mathrm{KL}'(X^{\hat{b}} \| Y^{\hat{b}}) = \mathrm{KL}(X^{\hat{b}} \| Y^{\hat{b}}) + \frac{1}{4} \mathbb{E}[\log(\rho_1^{\hat{b}}(X_1^{\hat{b}})/\rho_1(X_1^{\hat{b}}))]
\tag{140}
$$

This additional term is proportional to the Kullback-Leibler divergence of $\rho_1^{\hat{b}}$ from the target PDF $\rho_1$. Using (139) as well as $\rho_1(x) = e^{-U_1(x) + F_1}$, we can now express (140) as

$$
\begin{aligned}
\mathrm{KL}'(X^{\hat{b}} \| Y^{\hat{b}}) &= \frac{1}{4} \int_0^1 \mathbb{E}\big[\epsilon_t |\nabla U_t(X_t^{\hat{b}})|^2 - \epsilon_t \Delta U_t(X_t^{\hat{b}}) - \nabla \cdot b_t(X_t^{\hat{b}})\big] dt \\
&\quad + \frac{1}{4} \mathbb{E}[\log \rho_0(X_0)] + \frac{1}{4} \mathbb{E}[U_1(X_1)] - \frac{1}{4} F_1.
\end{aligned}
\tag{141}
$$

This is Equation (24) in (Vargas et al., 2024) in which we set $\nabla \hat{\phi}_t(x) = \hat{b}_t(x)$ and we used that, for any $c_t : \mathbb{R}^d \to \mathbb{R}^d$, we have

$$
\mathbb{E} \int_0^1 c_t(X_t^{\hat{b}}) \cdot \overleftarrow{d} W_t = \sqrt{2\epsilon_t} \int_0^1 \mathbb{E}\big[\nabla \cdot c_t(X_t^{\hat{b}})\big] dt.
\tag{142}
$$

Note that we can neglect the term $\frac{1}{4} \mathbb{E}[\log \rho_0(X_0)]$ in (139) since it does not depend on $\hat{b}$, so that the minimization of (139) can be cast into the minimization of (after multiplication by 4)

$$
\begin{aligned}
&\int_0^1 \mathbb{E}\big[\epsilon_t |\nabla U_t(X_t^{\hat{b}})|^2 - \epsilon_t \Delta U_t(X_t^{\hat{b}}) - \nabla \cdot b_t(X_t^{\hat{b}})\big] + \mathbb{E}[U_1(X_1) - F_1] \\
&= \int_0^1 \int_{\mathbb{R}^d} \big[\epsilon_t |\nabla U_t(x)|^2 - \epsilon_t \Delta U_t(x) - \nabla \cdot b_t(x))\big] \rho_t^{\hat{b}}(x) dx + \int_{\mathbb{R}^d} (U_1(x) - F_1) \rho_1^{\hat{b}}(x) dx
\end{aligned}
\tag{143}
$$

where $\rho_t^{\hat{b}}$ solves (131).

Let us check that the minimizer of (143) is $\hat{b}_t = b_t$, the solution to (23), so that we also have $\rho_t^{\hat{b}} = \rho_t^b = \rho_t$. To this end, notice that the minimization of (143) can be performed with the method Lagrange multiplier, using the extended objective

$$
\begin{aligned}
&\int_0^1 \int_{\mathbb{R}^d} \left[\epsilon_t |\nabla U_t(x)|^2 - \epsilon_t \Delta U_t(x) - \nabla \cdot \hat{b}_t(x)\right] \rho_t^{\hat{b}}(x) dx dt + \int_{\mathbb{R}^d} (U_1(x) - F_1) \rho_1^{\hat{b}}(x) dx \\
&+ \int_0^1 \int_{\mathbb{R}^d} \lambda_t(x) \left(\partial_t \rho_t^{\hat{b}} - \epsilon_t \nabla \cdot (\nabla U_t \rho_t^{\hat{b}} + \nabla \rho_t^{\hat{b}}) + \nabla \cdot (\hat{b}_t \rho_t^{\hat{b}})\right) dx dt
\end{aligned}
\tag{144}
$$

where $\lambda_t(x)$ is the Lagrange multiplier to be determined. Taking the first variation of this objective over $\lambda$, $\rho^{\hat{b}}$, and $\hat{b}$, we arrive at the Euler-Lagrange equations

$$
\begin{aligned}
0 &= \partial_t \rho_t^{\hat{b}} - \epsilon_t \nabla \cdot (\nabla U_t \rho_t^{\hat{b}} + \nabla \rho_t^{\hat{b}}) + \nabla \cdot (\hat{b}_t \rho_t^{\hat{b}}), \qquad \rho_0^{\hat{b}} = \rho_0, \\
0 &= \epsilon_t |\nabla U_t|^2 - \epsilon_t \Delta U_t - \nabla \cdot \hat{b}_t \\
&\quad - \partial_t \lambda_t + \epsilon_t \nabla U_t \cdot \nabla \lambda_t - \epsilon_t \Delta \lambda_t - \hat{b}_t \cdot \nabla \lambda_t \qquad \lambda_1 = -U_1 + F_1 \\
0 &= \nabla \rho_t^{\hat{b}} - \rho_t^{\hat{b}} \nabla \lambda_t
\end{aligned}
\tag{145}
$$

We can check that $\hat{b}_t(x) = b_t(x)$, $\rho_t^{\hat{b}}(x) = \rho_t(x) = e^{-U_t(x) + F_t}$, and $\lambda_t(x) = -U_t(x) + F_t$ is a solution: indeed this solves the first and the last equations in (145) and reduces the second to

$$
\begin{aligned}
0 &= \left[\epsilon_t |\nabla U_t|^2 - \epsilon_t \Delta U_t - \nabla \cdot b_t\right] \\
&\quad + \partial_t U_t - \partial_t F_t - \epsilon_t |\nabla U_t|^2 + \Delta U_t + \nabla b_t \cdot \nabla U_t \\
&= -\nabla \cdot b_t + \partial_t U_t - \partial_t F_t + \nabla b_t \cdot \nabla U_t
\end{aligned}
\tag{146}
$$

which is satisfied since $b_t$ solves (23).

# G. Hutchinson's Trace Estimator for the Evaluation of $\nabla \cdot \hat{b}_t(x)$

It is well-known that, if $\nabla \nabla b_t(x)$ is bounded,

$$
\nabla \cdot \hat{b}_t(x) = \frac{1}{2\delta} \mathbb{E}\left[\eta \cdot \left(\hat{b}_t(x + \delta \eta) - \hat{b}_t(x - \delta \eta)\right)\right] + O(\delta^2),
\tag{147}
$$

where $0 < \delta \ll 1$ is an adjustable parameter and $\eta \sim N(0, \mathrm{Id})$. Indeed we have

$$
\frac{1}{2\delta} \eta \cdot \left(\hat{b}_t(x + \delta \eta) - \hat{b}_t(x - \delta \eta)\right) = \eta^T \nabla b_t(x) \eta + O(\delta^2),
\tag{148}
$$

which implies (147) after taking the expectation over $\eta$.

We can use this formula to estimate the PINN loss via

$$
L_{\mathrm{PINN}}^{T,\delta}[\hat{b}, \hat{F}] = \int_0^T \mathbb{E}\left[R_t^\delta(x_t, \eta) R_t^\delta(x_t, \eta')\right] dt
\tag{149}
$$

where the expectation is now taken independent over $x_t \sim \hat{\rho}_t$, $\eta \sim N(0, \mathrm{Id})$, and $\eta' \sim N(0, \mathrm{Id})$, and we defined

$$
R_t^\delta(x, \eta) = \frac{1}{2\delta} \eta \cdot \left(\hat{b}_t(x + \delta \eta) - \hat{b}_t(x - \delta \eta)\right) - \nabla U_t(x) \cdot \hat{b}_t(x) - \partial_t U_t(x) + \partial_t \hat{F}_t
\tag{150}
$$

The expectation in (149) is unbiased since $\eta \perp \eta'$, and its accuracy can be controlled by lowering $\delta$.

# H. Details on Numerical Experiments

In the following we include details for reproducing the experiments presented in Section 3. An overview of the training procedure is given in Algorithm 1. Note that the SDE for the weights can replaced with (122) when learning with $\hat{\phi}_t$, as one would do with the action matching loss (119).

## H.1. Performance metrics

**Effective sample size.** We can compute the self-normalized ESS as

$$\text{ESS}_t = \frac{\left(N^{-1} \sum_{i=1}^{N} \exp\left(A_t^i\right)\right)^2}{N^{-1} \sum_{i=1}^{N} \exp\left(2A_t^i\right)} \tag{151}$$

at time $t$ along the SDE trajectory. We can use the ESS both as a quality metric and as a trigger for when to perform resampling of the walkers based on the weights, using, e.g. systematic resampling (Doucet et al., 2001; Bolić et al., 2004). Systematic resampling is one of many resampling techniques from particle filtering wherein some walkers are killed and some are duplicated based on their importance weights.

**2-Wasserstein distance.** The 2-Wasserstein distance reported in Table 1 were computed with 2000 samples from the model and the target density using the Python Optimal Transport library.

**Maximum Mean Discrepancy (MMD).** We use the MMD code from (Blessing et al., 2024) to benchmark the performance of NETS on Neal's funnel. We use the definition of the MMD as

$$\text{MMD}^2\left(\hat{\rho}, \rho\right) \approx \frac{1}{n(n-1)} \sum_{i,j}^{n} k\left(\hat{x}_i, \hat{x}_j\right) + \frac{1}{m(m-1)} \sum_{i,j}^{m} k\left(x_i, x_j\right) - \frac{2}{nm} \sum_{i}^{n} \sum_{j}^{m} k\left(\hat{x}_i, x_j\right) \tag{152}$$

where $\hat{x} \sim \hat{\rho}$ is from the model distribution and $x \sim \rho$ is from the target and $k : \mathbb{R}^d \times \mathbb{R}^d \to R$ is chosen to be the radial basis kernel with unit bandwidth.

## H.2. 40-mode GMM

The 40-mode GMM is defined with the mean vectors given as:

$$\mu_1 = (-0.2995,\ 21.4577), \qquad \mu_2 = (-32.9218,\ -29.4376),$$
$$\mu_3 = (-15.4062,\ 10.7263), \qquad \mu_4 = (-0.7925,\ 31.7156),$$
$$\mu_5 = (-3.5498,\ 10.5845), \qquad \mu_6 = (-12.0885,\ -7.8626),$$
$$\mu_7 = (-38.2139,\ -26.4913), \qquad \mu_8 = (-16.4889,\ 1.4817),$$
$$\mu_9 = (15.8134,\ 24.0009), \qquad \mu_{10} = (-27.1176,\ -17.4185),$$
$$\mu_{11} = (14.5287,\ 33.2155), \qquad \mu_{12} = (-8.2320,\ 29.9325),$$
$$\mu_{13} = (-6.4473,\ 4.2326), \qquad \mu_{14} = (36.2190,\ -37.1068),$$
$$\mu_{15} = (-25.1815,\ -10.1266), \qquad \mu_{16} = (-15.5920,\ 34.5600),$$
$$\mu_{17} = (-25.9272,\ -18.4133), \qquad \mu_{18} = (-27.9456,\ -37.4624),$$
$$\mu_{19} = (-23.3496,\ 34.3839), \qquad \mu_{20} = (17.8487,\ 19.3869),$$
$$\mu_{21} = (2.1037,\ -20.5073), \qquad \mu_{22} = (6.7674,\ -37.3478),$$
$$\mu_{23} = (-28.9026,\ -20.6212), \qquad \mu_{24} = (25.2375,\ 23.4529),$$
$$\mu_{25} = (-17.7398,\ -1.4433), \qquad \mu_{26} = (25.5824,\ 39.7653),$$
$$\mu_{27} = (15.8753,\ 5.4037), \qquad \mu_{28} = (26.8195,\ -23.5521),$$
$$\mu_{29} = (7.4538,\ -31.0122), \qquad \mu_{30} = (-27.7234,\ -20.6633),$$
$$\mu_{31} = (18.0989,\ 16.0864), \qquad \mu_{32} = (-23.6941,\ 12.0843),$$
$$\mu_{33} = (21.9589,\ -5.0487), \qquad \mu_{34} = (1.5273,\ 9.2682),$$
$$\mu_{35} = (24.8151,\ 38.4078), \qquad \mu_{36} = (-30.8249,\ -14.6588),$$
$$\mu_{37} = (15.7204,\ 33.1420), \qquad \mu_{38} = (34.8083,\ 35.2943),$$
$$\mu_{39} = (7.9606,\ -34.7833), \qquad \mu_{40} = (3.6797,\ -25.0242)$$

These means follow the definition given in the FAB (Midgley et al., 2023) code base that has been subsequently used in recent papers. The time-dependent potential $U_t(x)$ is given by the interpolation of means. To directly compare CMCD and NETS on this task, i.e. compare the log-variance path loss versus the PINN, we test the same mean-interpolation potential for CMCD using their code. The results are summarized in Table 3.

| | GMM-40 | | | |
| | ESS | | $\mathcal{W}_2$ | |
| Method | $n_{step} = 100$ | $n_{step} = 256$ | $n_{step} = 100$ | $n_{step} = 256$ |
| --- | --- | --- | --- | --- |
| NETS-PINN | $0.979 \pm 0.002$ | $0.988 \pm 0.001$ | $3.14 \pm 0.46$ | $3.01 \pm 0.49$ |
| CMCD-LV | $0.162 \pm 0.030$ | $0.51 \pm 0.04$ | $9.13 \pm 1.17$ | $3.62 \pm 0.55$ |

*Table 3.* Comparison of NETS and CMCD with the log-variance loss on GMM-40 using Effective Sample Size (ESS) and Wasserstein-2 distance for the mean-interpolation potential $U_t$.

### H.3. Neal's 10-$d$ funnel

The Neal's Funnel distribution is a 10-$d$ probability distribution defined as

$$x_0 \sim \mathsf{N}(0, \sigma^2), \quad x_{1:9} \sim \mathsf{N}(0, e^{x_0}) \tag{153}$$

where $\sigma = 3$ and we use subscripts here as a dimensional index and not as a time index like in the rest of the paper. Following this, we use as a definition of the interpolating potential:

$$U_t(x) = \frac{1}{2} x_0^2 (1 - t + \frac{t}{\sigma^2}) + \frac{1}{2} \sum_{i=1}^{d-1} e^{-tx_0} x_i^2 + (d-1)tx_0 \tag{154}$$

so that at $t = 0$, we have $U_0(x) = \frac{1}{2} x_0^2 + \frac{1}{2} \sum_{i=1}^{d-1} x_i^2$ and at time $t = 1$ we have the funnel potential given as $U_1(x) = \frac{1}{2\sigma^2} x_0 + \frac{1}{2} \sum_{i=1}^{d-1} e^{-x_0} x_i^2 + (d-1)x_0$.

### H.4. 50-$d$ Mixture of Student-T distributions

Following (Blessing et al., 2024), we use their mixture of 10 student-T distributions in 50 dimensions. We construct $U_t$ via interpolation of means from a single standard student-T distribution (mean 0). We use the same neural network as used in the GMM experiments.

To further drive home the fact that our annealed Langevin dynamics with transport can be taken post-training to the $\epsilon \to \infty$ limit to approach perfect sampling, we provide the following ablation from our model learned with the action matching loss given in Figure 4.

## I. Details of the $\varphi^4$ model

We consider the Euclidean scalar $\phi^4$ theory given by the action

$$S_{\text{Euc}}[\varphi] = \int \left[ \partial_\mu \varphi(x) \partial^\mu \varphi(x) + m^2 \varphi^2(x) + \lambda \varphi^4(x) \right] d^D x \tag{155}$$

where we use Einstein summation to denote the dot product with respect to the Euclidean metric and $D$ is the spacetime dimension. We are interested in acquiring a variant of this expression that provides a fast computational realization when put onto the lattice. Using Green's identity (integrating by parts) we note that

$$\int (\partial_\mu \varphi(x) \partial^\mu \varphi(x) d^d x = \int \partial_\mu \varphi \cdot \partial_\mu \varphi \, d^d x = - \int \varphi(x) \partial_\mu \partial^\mu \varphi(x) d^d x + \text{vanishing surface term} \tag{156}$$

so that

$$S_{\text{Euc}}[\varphi] = \int -\varphi(x) \partial_\mu \partial^\mu \varphi(x) + m^2 \varphi^2(x) + \lambda \varphi^4(x) \, d^d x. \tag{157}$$

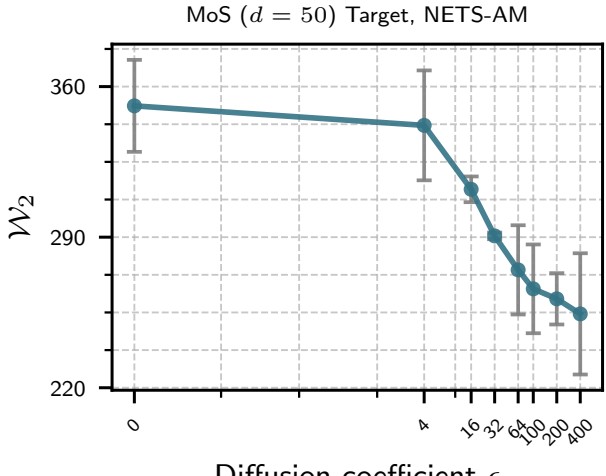

*Figure 4.* Reduction in $\mathcal{W}_2$ distance from taking the $\epsilon \to \infty$ limit in sampling with NETS. Note that the resolution of the SDE integration must increase to accommodate the higher stochasticity. Average taken over 3 sampling runs of 2000 walkers each.

Discretizing $S_{\text{Euc}}$ onto the lattice

$$\Lambda = \{a(n_0, \ldots, n_{d-1}) \mid n_i \in \{0, 1, 2, \ldots, L\}, i = 0, 1, \ldots, d, a \in \mathbb{R}_+\},$$

where $a$ is the lattice spacing used to define the physical point $x = an$, we use the forward difference operator to define

$$\partial_\mu \varphi(x) \to \tfrac{1}{a}[\varphi(x + \mu) - \varphi(x)] \qquad \partial_\mu \partial^\mu \varphi(x) \to \tfrac{1}{a^2}[\varphi(x + \mu) - 2\varphi(x) + \varphi(x - \mu)]. \tag{158}$$

Using these expressions, we write the discretized lattice action as

$$S_{\text{Lat}} = \sum_{x \in \Lambda} a^D \left[ \sum_{\mu=1}^{D} -\tfrac{1}{a^2}[\varphi_{x+\mu}\varphi_x - 2\varphi_x^2 + \varphi_{x-\mu}\varphi_x] + m^2\varphi_x^2 + \lambda\varphi_x^4 \right] \tag{159}$$

$$= \sum_x a^D \left[ 2Da^{-2}\varphi_x^2 - a^{-2}\sum_\mu [\varphi_{x+\mu}\varphi_x + \varphi_{x-\mu}\varphi_x] + m^2\varphi_x^2 + \lambda\varphi_x^4 \right] \tag{160}$$

$$= \sum_x a^d \left[ 2Da^{-2}\varphi_x^2 - 2a^{-2}\sum_\mu [\varphi_x\varphi_{x+\mu}] + m^2\varphi_x^2 + \lambda\varphi_x^4 \right] \tag{161}$$

$$= \sum_x a^D \left[ -2a^{-2}\sum_\mu \varphi_x\varphi_{x+\mu} + (2a^{-2}D + m^2)\varphi_x^2 + \lambda\varphi_x^4 \right] \tag{162}$$

where we have used the fact that on the lattice $\sum_x \varphi_x\varphi_{x+\hat{\mu}} = \sum_x \varphi_{x-\hat{\mu}}\varphi_x$ to get the third equality. It is useful to put the action in a form that is independent of the lattice spacing $a$. To do so, we introduce the re-scaled lattice field as

$$\varphi_x \to a^{D/2-1}\varphi_x, \quad m^2 \to a^2 m^2, \quad \text{and} \quad \lambda \to a^{4-D}\lambda. \tag{163}$$

Plugging these rescalings into (162) gives us the final expression

$$S_{\text{Lat}} = \sum_x \left[ -2\sum_\mu \varphi_x\varphi_{x+\mu} \right] + (2D + m^2)\varphi_x^2 + \lambda\varphi_x^4, \tag{164}$$

which we are to use in simulation.

**I.1. Free theory** $\lambda = 0$

Turning off the interaction makes it possible to analytically solve the theory. To do this, introduce the discrete Fourier transform relations

$$\varphi_k = \frac{1}{\sqrt{L^D}} \sum_x \varphi_x e^{-ik \cdot x} \tag{165}$$

$$\varphi_x = \frac{1}{\sqrt{L^D}} \sum_k \varphi_k e^{ik \cdot x} \tag{166}$$

for discrete wavenumbers $k = \frac{2l\pi}{L}$ with $l = 0, \cdots, L-1$. Plugging in (166) into the first part of (162), we get the expanded sum

$$\sum_x \left[ -2 \sum_\mu \hat{\varphi}_x \hat{\varphi}_{x+\mu} \right] \rightarrow -\frac{2}{L^d} \sum_x \sum_\mu \sum_k \sum_{k'} \varphi_k \varphi_{k'} e^{i(k+k') \cdot x} e^{ik' \cdot \mu} \tag{167}$$

$$= -2 \sum_\mu \sum_k \sum_{k'} \delta_{k,-k'} \varphi_k \varphi_{k'} e^{ik' \cdot \mu} \tag{168}$$

$$= -2 \sum_\mu \sum_k \varphi_k \varphi_{-k} e^{-ik_\mu} \tag{169}$$

$$= -2 \sum_\mu \sum_k \varphi_k \varphi_{k^*} e^{-ik_\mu} \tag{170}$$

$$= -2 \sum_\mu \sum_k |\varphi_k|^2 [\cos k_\mu + \cancel{i \sin k_\mu}] = -\sum_\mu \sum_k |\varphi_k|^2 \cos k_\mu \tag{171}$$

where $\phi^*$ indicates conjugation, and we got the first equality by the orthogonality of the Fourier modes, the second by the Kronecker delta, and the third by the reality of the scalar field. Proceeding similarly for the terms proportional to $\varphi^2$ gives us the expression

$$S_k = \sum_k \left[ m^2 + 2D - 2 \sum_\mu \cos k_\mu \right] |\varphi|^2 \tag{172}$$

The above equation can be written in quadratic form to highlight that the field may be sampled analytically

$$S_k = \frac{1}{L^d} \sum_k \varphi_k M_{k,-k} \varphi_{-k} \tag{173}$$

$$\text{where} \quad M_{k,-k} = \left[ m^2 + 2D - 2 \sum_\mu \cos k_\mu \right] \delta_{k,-k} \tag{174}$$

Note that this free theory can be sampled for any $m^2 > 0$.

**I.2. $\varphi^4$ numerical details**

We numerically realize the above lattice theory in D=2 spacetime dimensions. We use an interpolating potential with time dependent $m_t^2 = (1-t)m_0^2 + tm_1^2$, $\lambda_t = (1-t)\lambda_0 + t\lambda_1$ where $\lambda_0$ is always chosen to be 0 (though we note that you could run this sampler for any $U_0$ that you could sample from easily, not just analytically but also with existing MCMC methods). For the $L = 20$ (d $= L \times L = 400$ dimensional) experiments, we identify the critical point of the theory (where the lattices go from ordered to disordered) using HMC by studying the distribution of the magnetization of the field configurations as $M[\varphi^i(x)] = \sum_x \varphi^i(x)$, where summation is taken over all lattice sites on the $i^{th}$ lattice configuration. We identify this at $m_1^2 = -1.0$, $\lambda_1 = 0.9$ and use these as the target theory parameters on which to perform the sampling. For the $L = 16$ test (d $= 256$), we go past this phase transition into the ordered phase of the theory, which we identify via HMC simulations at $m_1^2 = -1.0$, $\lambda_1 = 0.8$.

