# OpenReview forum: "NETS: A Non-equilibrium Transport Sampler"
_ICML.cc/2025/Conference — ICML 2025 poster_

### Official Review · Reviewer_z8Lc · 2025-03-12

**Overall Recommendation:** 2

**Summary:**

The authors propose a method for sampling from unnormalized probability distributions. The method builds on diffusion-based sampling, where a learnable drift is added to the stochastic differential equation. The authors propose a PINN objective which allows for off-policy optimization and does not require differentiating through the simulations. Moreover, the authors addtionally propose an objective that is based on action matching. The methods are tested on a variety of sampling problems.

**Claims And Evidence:**

Claims are supported by theory. The reviewer has no concerns.

**Essential References Not Discussed:**

The authors did not cite related work that uses PINNs to sample from unnormalized densities, see [1,2]. Moreover, one of the seminal papers [3] on diffusion-based sampling was not cited.

Another paper that might be of interest for the authors is [4] which also builds on dynamic measure transport with resampling schemes

[1] Shi, Zhekun, et al. "Diffusion-PINN Sampler." arXiv preprint arXiv:2410.15336 (2024).
[2] Sun, Jingtong, et al. "Dynamical measure transport and neural PDE solvers for sampling." arXiv preprint arXiv:2407.07873 (2024).
[3] Richter, Lorenz, and Julius Berner. "Improved sampling via learned diffusions." arXiv preprint arXiv:2307.01198 (2023).
[4] Chen, Junhua, et al. "Sequential controlled langevin diffusions." ICLR 25

**Experimental Designs Or Analyses:**

The comparision between the different methods seems highly unfair. The authors use knowledge of the target density that significantly reduces the complexity of the problems. For instance, the method linearly interpolates the means of the 40 mode GMM with the prior potential, which effectively removes the difficuly of exploration. Moreover, similar handcrafted potentials are used for other targets as well.
 Are the authors using the same interpolation scheme for the baselines like FAB, or CMCD?

**Methods And Evaluation Criteria:**

The paper uses a variety of evaluation critera such as ESS, Wasserstein distance or MMD which fit for evaluating sampling methods. Moreover, the paper uses a variety of baselines, most of which are quite recent, which is also good.

**Other Comments Or Suggestions:**

A lot of the equations are stated but no context is provided which makes it difficult to read the paper.

**Other Strengths And Weaknesses:**

Strenghts:

- The paper proposes an off-policy objective which has potential to increase sample efficiency
- The methods avoids having to backpropagate though the simulation

Weaknesses:

- See Relation To Broader Scientific Literature and Experimental Designs Or Analyses

**Questions For Authors:**

- What is the difference between the proposed PINN objective and other related PINN based sampling methods, see e.g. [1,2]. The latter in particular also considers a prescribed density evolution.
- What insights does top row, left of Figure 3. give?
- The authors use CMCD-LV and CMCD-KL in Table 1. I assume this refers to the log-variance loss and the KL loss (the former is not mentioned cited). What is the benefit of the proposed method compared to CMCD-LV?

[1] Shi, Zhekun, et al. "Diffusion-PINN Sampler." arXiv preprint arXiv:2410.15336 (2024).
[2] Sun, Jingtong, et al. "Dynamical measure transport and neural PDE solvers for sampling." arXiv preprint arXiv:2407.07873 (2024).

**Relation To Broader Scientific Literature:**

It is not quite clear to the reviewer what the key contribution/novelty of this paper is. Sampling with PINNs was already done in [1,2]. The connection with Jarzynski’s equality was recently shown in [4] as noted by the authors. Combing diffusion-based sampling with SMC was also recently proposed in [4]. Moreover, off-policy learning with diffusion-sampler is possible using the log-variance loss, see [5].


[1] Shi, Zhekun, et al. "Diffusion-PINN Sampler." arXiv preprint arXiv:2410.15336 (2024).
[2] Sun, Jingtong, et al. "Dynamical measure transport and neural PDE solvers for sampling." arXiv preprint arXiv:2407.07873 (2024).
[3] Vargas, Francisco, et al. "Transport meets variational inference: Controlled monte carlo diffusions." arXiv preprint arXiv:2307.01050 (2023).
[4] Chen, Junhua, et al. "Sequential controlled langevin diffusions." ICLR 25
[5] Richter, Lorenz, and Julius Berner. "Improved sampling via learned diffusions." arXiv preprint arXiv:2307.01198 (2023).

**Theoretical Claims:**

The reviewer skimmed the Proofs in Appendix A which appear to be correct.

---

> ### Author Rebuttal · Authors · 2025-04-01
>
> We thank the reviewer for their careful reading of our paper and positive feedback. We are glad that you found the work well-written, theoretically sound, and that the numerical experiments demonstrate convincing evidence for our method's effectiveness. Below we address your comments and suggestions:
>
> **Experimental Designs Or Analyses:** The potential $U_t$ design is a feature we can exploit, similar to FAB or CMCD. In our drive link (https://tinyurl.com/netsicml) we compare with CMCD using identical interpolation, revealing CMCD struggles with this path: Using the same number of sampling steps as us, CMCD achieves only 15% ESS. To perform these experiments we worked with the authors to best benchmark their codebase.
>
> **Relation To Broader Scientific Literature** and **Essential References Not Discussed:** Thanks for pointing us to the papers by Shi, Zhekun, et al.; Sun, Jingtong, et al.; and Chen, Junhua, et al. We are happy to add a citation to these works, but we wish to note that they fall within the ICML policy of concurrent works: "https://icml.cc/Conferences/2025/ReviewerInstructions".
>
> We would also like to stress that, while variants of the PINN loss have appeared in the literature, our work makes the following new contributions:
>
> - Novel connection between the PINN loss and the Jarzynski weighting factors and annealed langevin dynamics. This result fits in naturally with physical interpretation of Jarzynski equality -- the PINN loss control on the variance of the Jarzynski weights (the dissipation) in the process connecting $\rho_0$ to $\rho_1$.
> - Show that we can better approach $\epsilon_t \to \infty$ limit in practice. Note that for our setup, perfect sampling is achieved in this limit, whether the learned transport is perfect or not. While this limit cannot be reached in practice without transport (as it would require taking astronomically large value of $\epsilon_t$ in general), we show that, with some learned transport added, even moderate values of $\epsilon_t$ can improve the sampling dramatically. This feature can be exploited after training as an explicit knob for tuning performance vs cost. This has not been recognized in other ML sampling literature nor in Vargas et al.
> - Show that the PINN loss directly controls the KL-divergence between the sampled and target distribution.
> - Directly characterize the minimizer of the action matching loss, which is also not known from previous work on it.
>
> Our method connects with CMCD, but we derive results through Fokker-Planck manipulations rather than Girsanov theorem.
>
> We'll add references to Richter and Berner's foundational work.
>
> **Other comments and suggestions:** Could you please specify which equations are stated without context? We will happily try to clarify in editing.
>
> **Questions for authors:**
>
> -*Difference between the proposed PINN objective and other related PINN based sampling methods:* See our reply above about the key new insights that our work provides.
> -*Insight of top row, left of Figure 3:* The top row (left) are example configurations of samples as we approach the critical point, and the bottom row are samples past the critical point, where they are fully magnetized. They illustrate the juxtaposition.
> -*Benefit of the proposed method compared to CMCD-LV:* We have included in the "additional experiments" section below results on CMCD-LV, which we worked with the CMCD authors to set up. Our results show that, unlike our approach, CMCD-LV exhibits instabilities if the number of sampling steps is too small in training: while the CMCD-LV loss is "off-policy" like our PINN loss, it still requires a trajectory on which to perform the optimization, and the generation of such a $n_{step}$ trajectory requires O($n_{step}$ $\times$ network size) memory. If the trajectory generation is performed over too few steps the CMCD LV loss becomes unstable. After discussing with CMCD authors, this is an issue generally when $n_{step} < 256$.
>
> **Additional experiments:**
> In the anonymous drive link https://tinyurl.com/netsicml:
>
> - We provide additional comparison between NETS and CMCD on the mean-interpolating time-dependent potential, as asked by another reviewer. We see that NETS performs well regardless of number of discretization steps, but the log-variance loss struggles for small steps and only performs well when $n_{step} = 256$ or more. We worked with the CMCD authors to implement their code and otherwise use their hyperparameters.
> - We have also set up a test on the Lennard Jones potential, and have included preliminary results. We are continuing to refine these experiments and will incorporate them in the final version.

---

### Official Review · Reviewer_5UR2 · 2025-03-14

**Overall Recommendation:** 4

**Summary:**

This paper introduces an algorithm for sampling from unnormalised probability distributions, through non-equilibrium sampling approaches.   When computing expectations with respect to the final-time marginal distribution, classical approaches to this would leverage AIS or equivalently Jarzynski / Crooks.      This approach, instead introduces a correction term, which corrects the discrepancy between the marginal distribution of the non-stationary 'Langevin' dynamics and the true distribution.    This discrepancy can be expressed  as a solution to a PDE, which is solved using a PINN-type loss.

The authors provide advice on tuning, suggestions for reducing cost of computations, and demonstrate the method on a Gaussian mixture model, Neal’s Funnel and Mixture of Student-T distributions, and statistical lattice field theory models.

**Claims And Evidence:**

The theoretical claims are supported by proofs in the Supplementary material.   The numerical experiments demonstrate the accuracy of the samples with respect to MMD and W2 distance, compared against a number of relevant comparable ML approaches to sampling from unnormalised distributions, which demonstrate convincing evidence that this method is achieves its claimed objective.

**Essential References Not Discussed:**

None.

**Experimental Designs Or Analyses:**

There are no experimental designs in this paper.

**Methods And Evaluation Criteria:**

The proposed approach is sensible, and scales well (compared to competing approaches) in high dimensions.   The evaluation criteria is both sensible and comprehensive.

**Other Comments Or Suggestions:**

Prop 2.3 Then ρt(x) is the PDF of X_t not X_T?

**Other Strengths And Weaknesses:**

Strengths:  The paper is well written, and the main arguments of the per are straightforward to follow.
Weakenesses:  While this is a good paper, the numerical experiments remain a bit lacking.  It would have been nice to see a wider range of potnential interactions.

**Questions For Authors:**

No further questions

**Relation To Broader Scientific Literature:**

This contribution sits in a wider body of literature which seek to correct dynamics by introducing KL-optimal control, building on ideas such as Follmer drift (e.g. [Huang et al, Convergence Analysis of Schrodinger-Follmer Sampler without Convexity, 2021], [Tzen and Raginski, 2019], [Vargas et al, Bayesian Learning via Neural Schrodinger-Follmer Flows, 2021]), and more recently [Vargas et al, Transport meets Variational Inference: Controlled Monte Carlo Diffusions, 2024].      The task of learning the vector field to correct the drift, as is done in this paper,  has been studied approached in various manners, [Reich, A dynamical systems framework for intermittent data assimilation, 2011], [Heng, Jeremy, Arnaud Doucet, and Yvo Pokern. Gibbs flow for approximate transport with applications to Bayesian computation. 2021] and earlier, [Vaikuntanathan & Jarzynski (2008)].

The approach has strong connections with CMCD [Vargas et al, 2024], which leverages a similar PINN loss (for an associated HJB equation) to this paper.   This is studied in detail in the supplementary information.

**Theoretical Claims:**

There are several theoretical claims made in the paper:
1. The Jarzynski inequality.
2. The fact that the corrected SDE (15) has exact marginals (prop 2.3)
3.  A weighted version, recovering [ Vaikuntanathan & Jarzynski (2008) ] prop 2.4
4. Derivation of the PINN objective (2.5)
5. KL controlled by PINN loss (2.6).
6. Discretised version of  Vaikuntanathan & Jarzynski (2008) (B1)
7. Connection with Feynman Kac (C1) for a specific form of b (gradient form).
8. D onwards - generalisations

All the proofs are sensible,

---

> ### Author Rebuttal · Authors · 2025-04-01
>
> We thank the reviewer for the positive feedback. We are glad that you found the contribution novel and the theory sound and thorough. Below we address your comments and suggestions and supply more information on experimental results:
>
>
> **Additional experiments**:
> In the anonymous drive link https://tinyurl.com/netsicml:
> - We provide additional comparison between NETS and CMCD on the mean-interpolating time-dependent potential, as asked by another reviewer. We see that NETS performs well regardless of number of discretization steps, but the log-variance loss struggles for small steps and only perform well when $n_{step} = 256$ or more. We worked with the CMCD authors to implement their code and otherwise use their hyperparameters.
> - We have also set up a test on the Lennard Jones potential, and have included preliminary results. We are continuing to refine these experiments and will incorporate them in the final version.

---

> > ### Comment · Reviewer_5UR2 · 2025-04-04
> >
> > Many thanks for your comments, I am happy with the proposed updates.  I will keep my score as-is.

---

### Official Review · Reviewer_kVWz · 2025-03-14

**Overall Recommendation:** 4

**Summary:**

The authors propose NETS, a Non-Equilibrium Transport Sampler that interpolates between two unnormalized densities $\rho_0$ and $\rho_1$ based on a user-defined choice of interpolant. A key contribution of the proposed approach is to introduce learning in the dynamics of continuous time annealed importance sampling by leveraging a learned drift which aims to minimize the variance of the importance weights, which can be combined with Sequential Monte Carlo by re-sampling during the simulation procedure based on the evolved importance weights. The paper introduces a physics informed neural networks (PINN) based loss to learn the drift function, and show that it additionally provides bounds on the KL divergence. Experiments are conducted on multiple different unnormalized densities, ranging from low and high dimensional Gaussian mixture models, Neal's funnel and mixture of Student-t distributions, as well as $\phi^4$ lattice theory and highlights the superiority of NETS over some of the established baselines.

**Claims And Evidence:**

The fundamental claims made by the work is proposing a framework for sampling according to an unnormalized density which uses learned methods and can be trained without back-propagating through the dynamics. The loss considered for this training is well motivated and the claims are supported by mostly clear theoretical evidence, with some concerns that I have raised in sections below. Broadly, I think the authors have provided good evidence (mostly theoretical) regarding the claims that they make and I outline more specific questions and concerns regarding the evidence provided in the later sections.

**Essential References Not Discussed:**

The authors cover most of the relevant literature off the top of my head.

**Experimental Designs Or Analyses:**

The experimental design and analyses conducted in this work is sound, except for one potential concern. The authors highlight that PIS did not converge on the Funnel task, while [1] shows that it does converge for this task. Could the authors clarify on what was the issue they came across?

[1] Sendera, Marcin, et al. "Improved off-policy training of diffusion samplers." Advances in Neural Information Processing Systems 37 (2024): 81016-81045.

**Methods And Evaluation Criteria:**

Yes, the methods and evaluation criteria considered in this work make sense. I do, however, have a few questions regarding the evaluation criteria and the benchmarks considered.

- In the proposed approach, is it possible to approximate the density $\hat{\rho}_1(x)$ corresponding to any sample $x$? In particular, I am wondering if it would be possible to additionally evaluate the proposed method to approximate the partition function $Z_1$?
- Is there a reason why the authors authors did not evaluate their models on the Lennard-Jones potential or more complex problems to better benchmark against related work?
- Why were some of the baselines (eg. iDEM, CMCD-LV) dropped for Table 2?
- Is there a reason why the authors do not consider SMC or HMC as one of the baselines in Tables 1,2?
- The authors should also consider evaluating against CMCD with the same interpolation of potentials as it is imperative to perform this fair comparison when evaluating the benefits of NETS.

**Other Comments Or Suggestions:**

- Did the authors forget to include a concluding section or is this by design? Currently, the end of the paper feels quite abrupt.

**Other Strengths And Weaknesses:**

**Strengths**

- The work is well motivated and tackles a challenging and very relevant problem. The combination of annealed importance sampling and diffusion style transport methods is novel and a good contribution to the field.
- The experiments highlight that the method seems to work well when compared to other learned samplers on the suite of tasks considered.
- It also provides some nice properties in the sense that one can compute expectations w.r.t the target measure through the use of importance weights.

**Weaknesses**

- I think the writing needs a bit more work. The way I understood it is that the authors consider defining an interpolation $\rho_t$ and then assume a drift $b$ which is trained with a loss directly derived from the continuity equation, i.e. $b$ is learned to satisfy the continuity equation for the marginals $\rho_t$. This learned drift is then artificially added to the Fokker Planck equation and then relevant terms are grouped together to define the drift, diffusion and importance sampling weight updates part of the coupled dynamics. In my opinion, this story could have been more clearly explained so that it makes it easier for a broader community to understand the work.
- The authors should consider a few more complex experimental settings, in particular the Lennard-Jones potential would be a good candidate.
- The authors mention that they can estimate the drift off-policy as well but they do not provide any experimentation with it. I think this would be a good ablation to add, and can be shown on perhaps a relatively easier task for some off-policy path.

**Questions For Authors:**

I just have one additional question, apart from the ones raised already.

- Within the related work section on augmenting sampling with learning, the authors talk about minimizing the KL divergence between model and target as well as stochastic optimal control. Aren't these two approaches the same, where the latter minimizes KL divergence between the considered path and a reference path measure through application of Girsanov's theorem?

Apart from this, I think the paper is well motivated and well positioned, so I will be happy to raise my score if the authors could clarify the questions that I have raised.

**Relation To Broader Scientific Literature:**

In my opinion, this is an interesting work and has key contributions to the scientific literature. In particular, it combines ideas from diffusion models and annealed importance sampling / sequential monte carlo and pushes the frontier of learned samplers.

**Theoretical Claims:**

I went over some of the theoretical claims made in the paper as well as some of the proofs outlined in Appendix A. While I enjoyed reading the theory outlined in the paper, there were some steps that were not obvious to me

- It was not clear as to how the authors jump from equation (6) to equation (7). In the latter, when talking about gradients and divergences, is it w.r.t the augmented state or the original state? It would be beneficial if the authors could actually sketch out the proof underlying this step.
- Assuming we have arrived at equation (7), how do the authors convert the partial differential equation to the coupled system of differential equation considered in Proposition 2.1? While the subsequent details and proof regarding expectation w.r.t the measure $\rho_t$ is clear, where and how does the couple dynamics come from the partial differential equation itself was not clear. I would request the authors to also put more focus and background into this step.
- Why is learning $\partial_t F_t$ a good idea? Is it because equation (49) implies that the objective is minimized if and only if the learned $F_t$ perfectly approximates the true free energy? But how does equation (48) lead to equation (49)? Are there any regularity conditions needed that $\rho_t$ vanishes faster than $\hat{b}_t$?
- In Section 2.5, can the authors talk about uniqueness of the solution that can be obtained? Why can there be more than one minimizer $(b, F)$?
- In Appendix A, within equation (35) when the authors use integration by parts, how do they ensure that the term at the boundary vanishes?

---

> ### Author Rebuttal · Authors · 2025-04-01
>
> We thank the reviewer for their valuable feedback, and we are happy to hear you found the work theoretically sound and novel. Below, we try to address all your questions, and provide some info on new additional experiments. We itemize our response according to the headings that appear in you review:
>
> **Methods And Evaluation Criteria:**
> - *estimating Z_1:* Yes, it's straightforward to construct an unbiased estimate if we know $Z_0$: $Z_1 = Z_0 \cdot \mathbb{E}[e^{A_1}]$. This follows from equation (21) at $t=1$,  which can beestimated empirically over trajectories from the coupled SDE/ODE (19,20).
> - *complexity of experiments:* We have included preliminary results on a Lennard-Jones system in an "Additional Experiments" section discussed below. Note that the $\phi^4$ models studied here are arguably more complex than the LJ-13 system: these $\phi^4$ models are either 256 or 400 dimensional problems, much bigger even than the LJ55 system, and are known as a proxy for one of the hardest sampling problems in research -- studying the strong force in lattice field theory. In particular, the $\phi^4$ models, like these lattice systems, suffer from what is known as critical slowing down in which MCMC algorithmic efficiency dramatically decays as one approaches the critical parameters. It has also been a target for neural samplers for some time [2].
> - *Dropped Baselines for iDEM, CMCD:* We only included baselines where we could either quote authors' results or work with them to verify implementation. We did the latter with CMCD on Funnel and GMM, but couldn't for MoS distribution.
> - *SMC/HMC*: SMC can be incorporated within our approach (this is what we refer to as "resampling"), but on its own isn't efficient enough for our sampling tasks. We tried HMC-type sampling (Inertial NETs, Appendix D.1) but saw little difference.
> - *CMCD interpolation path:* We thank the reviewer for pointing this out. We have included in the "additional experiments" section below results on CMCD on the path interpolating the means, which we worked with the CMCD authors to set up. With this path as well, CMCD-LV exhibits instabilities if the number of sampling steps is too small in training: while the CMCD-LV loss is "off-policy", it still requires a trajectory on which to perform the optimization, and the generation of such a $n_step$ trajectory requires O($n_{step}$ $\times$ network size) memory. If the trajectory generation is performed over too few steps the CMCD LV loss becomes unstable. After discussing with CMCD authors, this is an issue generally when $n_{step} < 256$. This will become apparent in the "additional experiments" section below.
>
> **Theoretical Claims:**
> - *eq (6) and (7):* SMC can be incorporated within our approach (as "resampling"), but on its own isn't efficient enough for our sampling tasks. We tried HMC-type sampling (Inertial NETs, Appendix D.1) but saw little difference.
> - *derivation of (7) and (10,11):* We'll add derivations in the appendix, as these aren't commonplace in ML literature.
> - *Why learn $\partial_t F$?:* The loss function is a PINN loss which means it is minimized at 0 (i.e., when the physics equation is solved). $\partial_t F_t$ is a free parameter in this equation and must be fit if one wants to learn off policy. We do indeed use that $\hat b_t \rho_t$ vanish at infinity, which is a consequence of the conservation of probability that prohibits the existence of a probability current at infinity.
> - *uniqueness of PINN minimizer:* The minimizer is unique *if and only if* the velocity field is learned in gradient form $\nabla \hat \phi_t = \hat b_t$.
> - *integration by parts:* This is again a consequence of conservation of probability requiring no probability flux at infinity.
>
> **Experimental Designs:**
> - *PIS not included:* The PIS authors confirmed to us that they used $\sigma^2 = 3$ (not $\sigma^2=9$ as stated in their paper). This makes the problem much easier, so we omitted this experiment.
>
>
> **Additional experiments**:
>
> In the anonymous drive link https://tinyurl.com/netsicml:
> - We provide a NETS vs CMCD comparison on mean-interpolating potentials which shows NETS performs well regardless of discretization steps, while log-variance struggles for small steps, improving only when $n_{step} ≥ 256$.
> - We've included preliminary Lennard Jones results and will incorporate refinements in the final version.
>
>
> We'll make your suggested expository edits when allowed to edit the paper. Thank you for your insights - if we've clarified everything, we'd appreciate an increase in your rating.
>
> [1] Alpha Collaboration, "Critical Slowing Down and Error Analysis in Lattice QCD simulations," Nuclear Physics B, 2010.
>
> [2] Albergo, Kanwar, Shanahan, "Flow-based Generative Models for MCMC in lattice field theory," Physical Review D, 2019.

---

> > ### Comment · Reviewer_kVWz · 2025-04-01
> >
> > Thanks to the authors for providing a detailed response. I am happy with the answers, and am looking forward to the updated manuscript with more details! I have also updated my rating accordingly.

---

### Official Review · Reviewer_VhJb · 2025-03-17

**Overall Recommendation:** 4

**Summary:**

This paper investigates sampling from a target distribution within the annealed importance sampling (AIS) framework. Inspired by Jarzynski equality, a continuous-time version of AIS can be formulated using an SDE for samples and an ODE for weights. Building on this, the paper proposes NETS by introducing an additional drift function into AIS, and propose learning this drift function using either a PINN loss or an action matching loss. Experimental results demonstrate the effectiveness of the proposed algorithm.

**Claims And Evidence:**

Yes.

**Essential References Not Discussed:**

Yes. One related reference is missing: Junhua Chen et al. Sequential Controlled Langevin Diffusions. ICLR, 2025.

**Experimental Designs Or Analyses:**

Yes.

**Methods And Evaluation Criteria:**

Yes.

**Other Comments Or Suggestions:**

- 2.2. Non-equilibrium sampling with importance weights: I didn't initially see how the right-hand side of Eq. (6) could be related to weights until the introduction of Jarzynski equality. The readability could be improved.

- Line 669: the term $\hat{b}_t$ is missing in Eq.(40). Should it be $\nabla U_t \cdot \hat{b}_t$? It seems that this omission also occurs in other equations in the appendix. Please check it.

**Other Strengths And Weaknesses:**

Strengths:

- The paper is generally well-structured and theoretically sound.

- A novel objective based on PINN or action matching is proposed to learn an additional drift function in the context of AIS.

Weaknesses:

- The main proposition of the paper (Proposition 2.4) has already been presented in Vargas et al. (2024), although the current derivations are arguably simpler to some extent.

- Please see the questions below for the authors.

---

Reference:

Francisco Vargas et al. Transport meets Variational Inference: Controlled Monte Carlo Diffusions. ICLR, 2024.

Jingtong Sun et al. Dynamical Measure Transport and Neural PDE Solvers for Sampling. Arxiv, 2024.

**Questions For Authors:**

- I have the following questions when comparing NETS with CMCD:

  - Regarding the sentence 'requiring either backpropagation through the SDE or computation with a numerically unstable reference measure on a fixed grid' (Line 73), I guess that the first drawback can be addressed by off-policy divergences, e.g., log-variance. Could you clarify what is meant by 'computation with a numerically unstable reference measure on a fixed grid'? Should the grids be tuned to avoid instability?

  - Regarding the sentence 'Moreover, our optimize-then-discretize framework allows for post-training adaptation of both step size and time-dependent diffusion, providing tunable parameters to enhance performance.'  (Line 79), I am curious if this approach can be applied to other diffusion-based samplers, or what might prevent other samplers from doing the same?

  - Aside from the fact that the PINN loss can be trained in an off-policy manner, are there any other advantages over KL divergence optimization? Additionally, have you considered experimenting with CMCD-LV, which can also be trained in an off-policy manner?

  - Missing reference: Junhua Chen et al. Sequential Controlled Langevin Diffusions. ICLR, 2025, where CMCD combined with SMC (AIS with resampling) is proposed. I am courious about the comparision in terms of both concept and experiment.

- Figure 4: Taking $\epsilon_t \rightarrow \infty$ shows improved performance, although with more discretization steps. This should be the motivation for introducing an additional drift function - it may be imperfect for most of the time but still helpful, thus leading to fewer discretization steps?

- Line 614 - 628: Is the definition of $g_t$ sepicial? Usually, we don't need $e^a$ to define the marginal distribution. The motivation here is to link it to the weight function? If so, does Eq.(36) describe the evolution of a *weighted* distribution?


---

Update after rebuttal: I have raised my score to 4.

---

**Relation To Broader Scientific Literature:**

This paper introduces a novel sampling algorithm inspired by concepts from nonequilibrium physics, aimed at enabling efficient sampling from unnormalized distributions.

**Theoretical Claims:**

Yes.

---

> ### Author Rebuttal · Authors · 2025-04-01
>
> We thank the reviewer for their valuable feedback, and we are happy to hear you found the work theoretically sound and novel. Below,  we try to address all your questions, and provide some info on new additional experiments. We itemize our response according to the headings that appear in you review:
>
> **References**
> - Thanks for pointing us to this paper. The SMC that they propose is also something that we propose in this work (when we may choose to do 'resampling'). We are happy to add a citation to this work, but we wish to note that this falls within the ICML policy of concurrent works:  "https://icml.cc/Conferences/2025/ReviewerInstructions".
>
>
> **Weaknesses**
> - Proposition 2.4 in general is not new, though this derivation is. Please note that it is also not new in Vargas et al. The essential machinery of this proposition has been known for 20 years in the statistical physics community (Jarzynski 1999, Vaikuntanathan & Jarzynski 2008), and neither us nor Vargas et al. are claiming to have discovered this equality. However, we both provide different derivations of it that make it more interpretable for sampling. Vargas et al. prove this result through the use of Girsanov, and here we provide a proof of it through simple manipulations of the the Fokker-Planck equation and other PDEs.
>
>
> **Other Comments Or Suggestions:**
> - We are happy to improve the readability of equation 6 when we can edit the text if the paper is accepted.
> - thanks for catching the typo in Eq (40).
>
> **Questions:**
> - *backprop thru sde and numerical instability:* The first drawback cannot be entirely addressed by the CMCD log-variance loss. It is still a loss function that must use samples along a *trajectory* as input, and the generation of such a trajectory of lengthn $n_{step}$ requires $O$($n_{step}$ $\times$ network size) memory. By numerical instability, we mean that the CMCD LV loss becomes unstable if the trajectory generation is performed over too few steps. After discussing with CMCD authors, this is an issue generally when $n_{step} < 256$. This will become apparent in the "additional experiments" section below. Let us also stress that, in contrsat, our PINN loss can be evaluated pointwise in space and time.
> - *Optimize then discretize:* Thanks for the observation. We imagine most other diffusion models could do this as well, but the main obstacle would be whether or not their method allows for an adaptable diffusion coefficient *post-traing* (like our method). If not, increasing the diffusion would require decreasing the time-step for trajectory generation during training.
> - *PINN vs KL*: The benefits of the off-policy nature of the PINN loss manifest themselves in two ways: 1) The PINN loss is valid with respect to any sampling distribution), and 2) if you want to use simulated samples from your model, you do not need to backpropagate through their generation. The KL loss explicitly needs samples from the model distribution, and requires a backprop through the solve. As mentioned earlier, the log-var loss still needs to generate trajectories but has a smaller memory footprint. We did test the log-variance more, too. See: "additional experiments".
> - *Paper by Chen et al:* Thanks again for pointing us to this. We are happy to reference it, though again per ICML policy it is out of the scope of this submission to benchmark it.
> - *$\epsilon \rightarrow \infty$ limit:*   Yes, that is exactly the motivation of the additional drift function. For our setup, perfect sampling is achieved in this limit, whether the learned transport is perfect or not. While this limit cannot be reached in practice without transport (as it would require taking astronomically large value of $\epsilon_t$ in general), we show that, with some learned transport added, even moderate values of $\epsilon_t$ can improve the sampling dramatically. This feature can be exploited after training as an explicit knob for tuning performance vs cost. This has not been recognized in other ML sampling literature.
>
>
> **Additional experiments**:
> In the anonymous drive link https://tinyurl.com/netsicml :
> - We provide additional comparison between NETS and CMCD on the mean-interpolating time-dependent potential, as asked by another reviewer. We see that NETS performs well regardless of number of discretization steps, but the log-variance struggles for small steps. Note that LV begins to perform better when $n_{step} = 256$. We worked with the CMCD authors to implement their code and otherwise use their hyperparameters. This is the numerical instability we referred to earlier.
> - We have also set up a test on the Lennard Jones potential, and have included preliminary results. We are continuing to refine these experiments and will incorporate them in the final version.
>
> Please let us know if you have any other questions, and thanks again for your insights. If we have clarified everything for you, we would greatly appreciate an increase in your rating.

---

> > ### Comment · Reviewer_VhJb · 2025-04-02
> >
> > I appreciate the authors' time and effort in addressing my concerns and conducting additional experiments.
> >
> > I am satisfied with the responses. However, the authors did not seem to answer the question posted previously:
> >
> > - Line 614 - 628: Is the definition of $g_t$ sepicial? Usually, we don't need $e^a$ to define the marginal distribution. The motivation here is to link it to the weight function? If so, does Eq.(36) describe the evolution of a *weighted* distribution?
> >
> > During the rebuttal period, I have a few follow-up questions regarding the methodology:
> >
> > - You correctly pointed out that optimizing the LV loss requires simulating entire trajectories. I have not yet played with NETS and PINN loss. Based on Algorithm 1 in the paper, we should also simulate entire trajectories when optimizing the PINN loss? Additional experimental results do indicate that the PINN loss remains stable when $n < 256$.
> >
> > - Given that different number of discretization steps can be used during inference, does $\epsilon_{t}$ remain fixed? $\epsilon_{t} \Delta t$ can be coupled together and $\Delta t$ is changed, allowing for a changable diffusion coffiecient. Also, have the authors tested the scenario, for example, training NETS with fewer discretization steps, while evaluating it with a fixed, but relatively more number of steps?
> >
> > - Regarding KL control, since we usually optimize a path-wise objective, have the authors considered the following relation: $\log \frac{\mathrm{d} \overrightarrow{\mathbb{P}}}{\mathrm{d} \overleftarrow{\mathbb{P}}} = \int_{0}^{T} \left( - \nabla \cdot \widehat{b}_t + \nabla U_t \cdot \widehat{b}_t + \partial_t U_t - \partial_t \widehat{F}_t \right) \mathrm{d} t$?

---

> > > ### Author Response · Authors · 2025-04-02
> > >
> > > Thank you for these additional comments and sorry for missing your question about $g_t$.
> > >
> > > **Regarding the function $g_t(x) = \int_{\mathbb R} e^a f_t(x,a) da$:** it is *not* the  marginalization over $x$ of the extended probability density $f_t(x,a)$ (which would indeed read $\int_{\mathbb R} f_t(x,a) da$ *without* the factor $e^a$, as you point out) but rather the unnormalized density given explicitly by:
> > > $$
> > > g_t(x) = Z_0^{-1} e^{-U_t(x)}
> > > $$
> > > This is what is established in Eq. (37) and it implies that
> > > $$
> > >  \int_{\mathbb R^d} g_t(x) dx = Z_tZ_0^{-1}
> > > $$
> > > and, given any test function $\phi$,
> > > $$
> > > \int_{\mathbb R^d} \phi(x) g_t(x) dx = Z_0^{-1} \int_{\mathbb R^d} \phi(x) e^{-U_t(x)} dx
> > > $$
> > > Since $g_t(x) = \int_{\mathbb R} e^a f_t(x,a) da$, these equations can also be written as
> > > $$
> > >  Z_tZ_0^{-1}  = \int_{\mathbb R^{d+1}} e^a f_t(x,a) dx da \equiv \mathbb{E}[e^{A_t}]
> > > $$
> > > and
> > > $$
> > > Z_0^{-1}\int_{\mathbb R^d} \phi(x) e^{-U_t(x)} dx =  Z_0^{-1} \int_{\mathbb R^{d+1}} e^a \phi(x) f_t(x,a) dx da \equiv Z_0^{-1}\mathbb{E}[e^{A_t}\phi(X_t)]
> > > $$
> > > Dividing the second equation by the first establishes that
> > > $$
> > > \frac{\mathbb{E}[e^{A_t}\phi(X_t)]}{\mathbb{E}[e^{A_t}]} = Z_t^{-1}\int_{\mathbb R^d} \phi(x) e^{-U_t(x)} dx
> > > $$
> > > which is the result of Proposition 2.1.
> > >
> > > We will be happy to clarify the role and meaning of $g_t(x)$ in the revised version.
> > >
> > >
> > > Regarding your **other questions**:
> > >
> > > - In Algorithm 1 in the paper, we specify a trajectory, but it is merely to get any samples over $[0,T] \times \mathbb{R}^d$, not to backpropagate through trajectories. The vector field $b_t$ simply needs to be fit to the PINN using the sampled data, so the stability of the loss is not influenced by the trajectory. We can store these samples in a replay buffer and draw a mini-batch to learn over. As you say, with LV-CMCD, the loss involves the trajectories themselves. In contrast, with NETS the trajectories only play a role of getting decent samples to evaluate the PINN loss on.
> > > - In general, we can play around with $\epsilon_t$ but to keep discretization bias low we need to scale $dt$ down with greater $\epsilon_t$. For a fixed $dt$, it is probably best to use the largest $\epsilon_t$ that one can given the discretization (the benefit of which is captured in Figure 3). Note also that a discrete time version of the sampling and weight computation is given in the appendix that addresses discretization error.
> > > - The Crooks equation you point out could indeed be used to derive our KL bound: to remain consistent with the general approach we take in the paper, we provide an alternative proof, based on using the FPE. We could make this connection with the path KL (a bit like what we do when we make a connection with CMCD) if you think that it is useful.
> > >
> > > Please let us know if this addresses all of your questions, and thanks again for your insights.

---

### Decision · Program_Chairs · 2025-05-01

**Decision:**

Accept (poster)

**Comment:**

This paper introduces NETS, a non-equilibrium transport sampler that extends annealed importance sampling (AIS) by learning an adaptive drift term using physics-informed neural networks (PINNs) or action-matching losses. Inspired by Jarzynski’s equality, the approach aims to reduce the variance of importance weights. The method builds connections to recent advances in controlled Langevin dynamics and is evaluated on several benchmark distributions, including high-dimensional Gaussian mixtures and lattice field theory models.

The work is grounded in theoretical foundations and addresses a timely challenge in probabilistic inference. Reviewers particularly appreciated the novel connection between the PINN loss and the Jarzynski weighting factors. Experimental results on synthetic and structured benchmarks show promising improvements over established baselines.

One concern raised by reviewers is the overlap with existing work on diffusion-based samplers and PINN-based training. The authors provided a detailed and thoughtful rebuttal clarifying their contributions and distinctions.

Overall, this paper offers technically sound contributions and proposes a compelling new direction for improving sampling efficiency. The connection between the PINN loss and Jarzynski’s equality is particularly insightful. Most reviewers are positive, and the remaining concerns appear addressable in revision.